# Neuroblast-specific open chromatin allows the temporal transcription factor, Hunchback, to bind neuroblast-specific loci

**Sonia Q Sen[1], Sachin Chanchani[1], Tony D Southall[2], Chris Q Doe[1]***

[1]Institute of Neuroscience, Institute of Molecular Biology, Howard Hughes Medical Institute, University of Oregon, Eugene, United States; [2]Department of Life Sciences, Imperial College London, London, United Kingdom

**Abstract** Spatial and temporal cues are required to specify neuronal diversity, but how these cues are integrated in neural progenitors remains unknown. *Drosophila* progenitors (neuroblasts) are a good model: they are individually identifiable with relevant spatial and temporal transcription factors known. Here we test whether spatial/temporal factors act independently or sequentially in neuroblasts. We used Targeted DamID to identify genomic binding sites of the Hunchback temporal factor in two neuroblasts (NB5-6 and NB7-4) that make different progeny. Hunchback targets were different in each neuroblast, ruling out the independent specification model. Moreover, each neuroblast had distinct open chromatin domains, which correlated with differential Hb-bound loci in each neuroblast. Importantly, the Gsb/Pax3 spatial factor, expressed in NB5-6 but not NB7-4, had genomic binding sites correlated with open chromatin in NB5-6, but not NB7-4. Our data support a model in which early-acting spatial factors like Gsb establish neuroblast-specific open chromatin domains, leading to neuroblast-specific temporal factor binding and the production of different neurons in each neuroblast lineage.
DOI: https://doi.org/10.7554/eLife.44036.001

***For correspondence:**
cdoe@uoregon.edu

**Competing interests:** The authors declare that no competing interests exist.

## Introduction

The generation of neuronal diversity in mammals and Drosophila is a multi-step process. The initial step is the production of the neuroectoderm (ventral in Drosophila, dorsal in mammals) that gives rise to neural progenitors. In both systems, the neuroectoderm and neural progenitor population acquire regional differences due to the action of Hox genes and spatial patterning genes (*Jessell, 2000*). Although spatial patterning generates diversity within the neural progenitor population, it is insufficient to account for the neuronal diversity in the mature nervous system. Expanding neural diversity requires a second step called temporal patterning, where individual neural progenitors produce a sequence of distinct neurons and glia (*Doe, 2017*). In both Drosophila and mammals, this process appears to be regulated, in part, by temporal transcription factors (TTFs) that are sequentially expressed within individual neural progenitors (*Kohwi and Doe, 2013*). Although a great deal is known about how spatial factors generate regional diversity, and much has recently been learned about temporal patterning mechanisms, virtually nothing is known about how spatial factors and TTFs are integrated to specify distinct neuronal identities in spatially distinct progenitor populations.

Drosophila is an excellent model system to investigate how spatial and temporal factors are integrated during neurogenesis, due to a deep understanding of neural progenitor (neuroblast) lineages, and the molecular mechanisms involved in both spatial and temporal patterning during neurogenesis. The Drosophila neuroectoderm produces a bilateral array of 30 neuroblasts in each

**eLife digest** The human brain is considered to be the most complicated object in the universe, but it only takes a handful of stem cells to make one. The process depends on two types of information: signals separated across space and time. Spatial cues tell a stem cell what type of cell it is going to be, while temporal cues work as molecular clocks to generate a sequence of different neurons over time. Together, these cues generate the large array of cell types in the nervous system.

Each stem cell occupies its own space in the developing body and receives its own spatial cues, but they all follow the same timeline. For example, proteins called transcription factors act as molecular clocks and interact with specific genes, telling the cell when to turn them on or off. The same series of transcription factors operates in different stem cells, but they have different effects. So far, it has been unclear whether spatial and temporal signals work independently or sequentially to generate new cell types.

To find out, Sen et al. studied two distinct, developing stem cells in fruit flies, which receive different spatial signals. Transcription factors only work if they are able to get to their target genes. Cells can open or close access to different genes by changing the structure of the chromatin wrapping that surrounds the genes. In the experiments, a marker was used to reveal the areas of open chromatin in each of the cells. Another marker was used to track the transcription factors. The results showed that the areas of open chromatin varied between stem cells. Moreover, although both cells used the same transcription factor called Hunchback, it targeted different genes in each stem cell. This was due to changes in the chromatin wrapping: Hunchback only acted in areas where the chromatin was open. This suggests that the spatial cues first sculpt the chromatin, making some genes easier to get to than others. Then, the same transcription factors go to the accessible gene, which will differ from one stem cell to another.

These findings help us to understand how different types of brain cells develop, which may also aid us in finding a way how to engineer specific cell types. If we could turn stem cells into different types of brain cells, it might help us to treat brain diseases. This may involve giving the right spatial signal before starting the temporal cues.

DOI: https://doi.org/10.7554/eLife.44036.002

segment, named according to their row and columnar position within the two dimensional neuroblast array (*Figure 1A*, left). Each neuroblast has a unique identity based on its distinct molecular profile and each neuroblast produces a unique and stereotyped family of neurons.

Spatial patterning factors that specify neuroblast identity have been characterized, and all of them are transcription factors or signalling pathways with transcription factor effectors. Henceforth we refer to these spatial factors as 'spatial transcription factors' or STFs, paralleling the naming of temporal transcription factors as TTFs. The Gooseberry (Gsb) Pax-3 family transcription factor is expressed in row 5 neuroblasts; loss of Gsb transforms row 5 neuroblasts into row 3/4 identity, and misexpression of Gsb transforms row 3/4 neuroblasts into row 5 identity. Importantly, transient misexpression of Gsb in the neuroectoderm, prior to neuroblast formation, is sufficient to generate ectopic row 5 neuroblasts, suggesting that neuroblast identity is determined in the neuroectoderm and maintained during the subsequent neuroblast lineage (*Skeath et al., 1995*; *Bhat, 1996*). Thus, Gsb is one of the best characterized STFs. Similarly, the secreted Wingless (Wg) protein is produced by row 5 neuroectoderm, where it is required to specify the adjacent row 4 and 6 neuroblast identity that is maintained in the row 4 and 6 neuroblasts (*Chu-LaGraff and Doe, 1993*). Precise inactivation of a temperature-sensitive Wg protein showed that loss of Wg activity in the neuroectoderm resulted in loss of neuroblast identity, whereas inactivation of Wg after neuroblast formation had no effect, showing that transient Wg generates row 4 and 6 neuroblast identity (*Chu-LaGraff and Doe, 1993*). In addition, Hedgehog (Hh) expression in row 6/7 neuroectoderm is required to specify neuroblast identity in adjacent rows 1/2 (*McDonald and Doe, 1997*). Finally, Engrailed expression in the neuroectoderm is required for the proper development of row 6/7 neuroblasts, and transient Engrailed misexpression generates ectopic row 7 neuroblast identity (*Deshpande et al., 2001*).

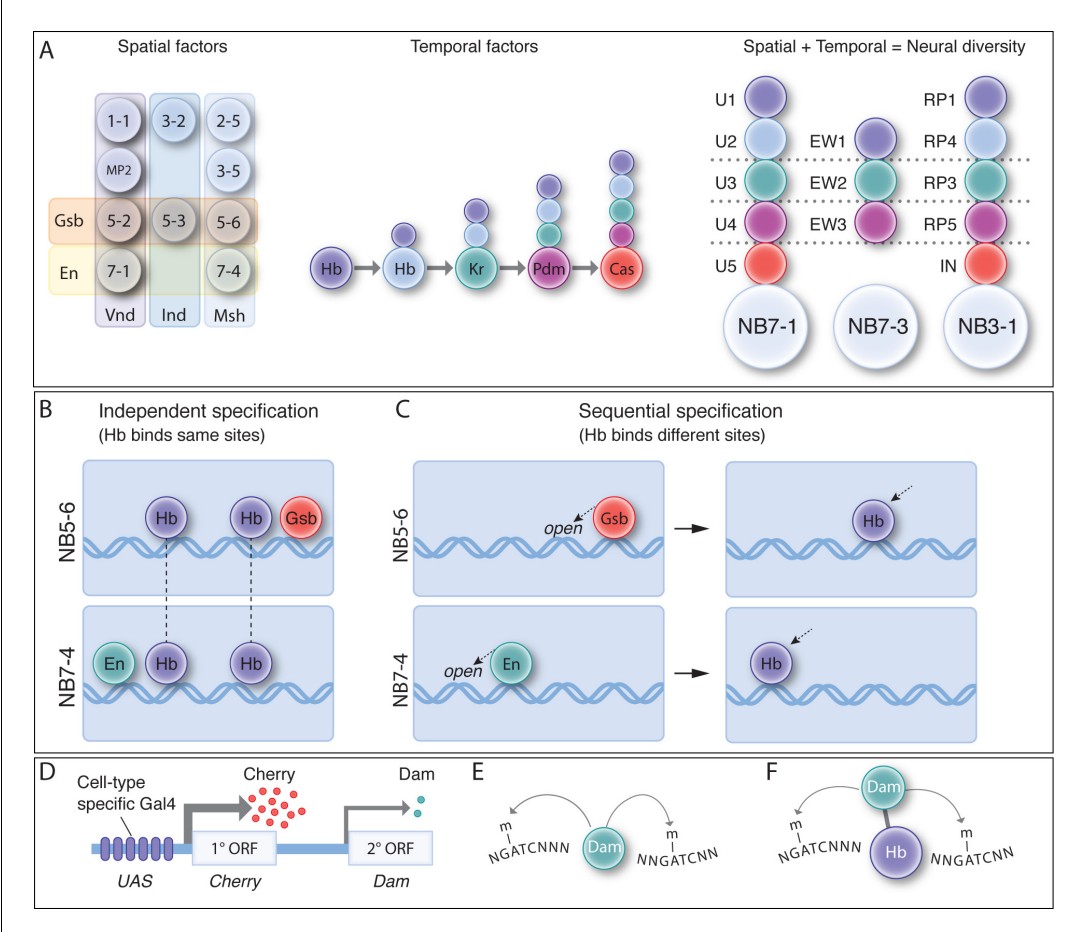

**Figure 1.** Spatial and temporal cues are integrated to generate neuronal diversity. (**A**) Spatial and temporal patterning. (Left) As neuroblasts delaminate from the neuroectoderm, they experience spatial transcription factors (e.g. Gsb, En, Vnd, Ind, Msh shown) that gives each neuroblast a unique molecular identity. (Middle) TTFs are sequentially expressed in most neuroblasts to specify GMC/neuronal identity based on birth-order. (Right) The integration of spatial and temporal factors specifies lineage-specific neuronal identity. (**B**) Independent specification: in this hypothesis, STFs and TTFs bind genomic targets independently, and their combinatorial effect specifies distinct neuroblast identity. In this model, TTF targets are the same in different NBs. (**C**) Sequential specification: in this hypothesis, STFs act first to bias or restrict subsequent TTF genomic binding, leading to the production of different neurons from different neuroblasts. In this model, TTF targets are the different in different NBs (**D–F**) The TaDa and CaTaDa Materials and method. See text for details.

DOI: https://doi.org/10.7554/eLife.44036.003

Taken together, these spatial patterning experiments show that neuroblast spatial identity is specified in the neuroectoderm by the transient action of STFs expressed in different neuroblast rows.

Spatial patterning does not only generate distinct rows of neuroblasts, but also distinct neuroblast columns. During the first stages of neuroblast formation there are three distinct columns of neuroblasts, each specified by a conserved homeodomain protein. Vnd is expressed in a medial column of neuroectoderm, Ind is expressed in an intermediate column, and Msh (Flybase: Drop) is expressed in the lateral column (*Figure 1A*, left) (*Isshiki et al., 1997*; *McDonald et al., 1998*; *Weiss et al., 1998*). Loss of function and misexpression studies show that each is necessary and partially sufficient for specifying columnar neuroblast identity (*Isshiki et al., 1997*; *McDonald et al., 1998*; *Weiss et al., 1998*). It is likely that these columnar factors function in the neuroectoderm, like spatial row factors, because they do not persist throughout neuroblast lineages. All three of these STFs have conserved mammalian orthologs with similar medial-lateral expression in the neuroectoderm (*Weiss et al., 1998*). Overall, the combination of row and columnar STFs are likely to generate the observed 30 distinct neuroblast identities. Hox factors provide an additional spatial cue that distinguishes segmental differences in neuroblast identity (*Prokop and Technau, 1994*).

Whereas spatial patterning generates 30 different neuroblast identities, temporal patterning is required to generate different progeny within each neuroblast lineage. Most neuroblasts sequentially express a series of four TTFs as they divide to generate ganglion mother cell (GMC) progeny, and the specific TTF inherited by each GMC determines its identity (*Kohwi and Doe, 2013*; *Li et al., 2013*; *Doe, 2017*). Embryonic ventral nerve cord (VNC) neuroblasts undergo a TTF cascade that progresses from Hunchback (Hb; Ikaros zinc finger family) to Krüppel (zinc finger family) to the redundant Nubbin/Pdm2 (Pdm) to Castor (Cas; Casz1 zinc finger family) (*Figure 1A*, middle). Other neuroblasts in the larval VNC, brain, and optic lobes undergo a similar TTF cascade to increase neuronal diversity, although the identity of the TTFs differs in each region (*Li et al., 2013*; *Doe, 2017*). The Hb-Kr-Pdm-Cas TTF cascade has been particularly well-characterized, with each factor being necessary and sufficient to specify the neuronal identity produced during its window of expression (*Isshiki et al., 2001*; *Novotny et al., 2002*; *Kanai et al., 2005*; *Grosskortenhaus et al., 2006*; *Tran and Doe, 2008*; *Kohwi et al., 2013*). Importantly, each TTF specifies a different type of neuron in each neuroblast lineage, showing that spatial identity provides a different context for Hb function in each neuroblast (*Figure 1A*, right). Understanding this 'context' at a mechanistic level is the goal of our experiments below.

The role of TTFs is best exemplified by Hb, the first TTF in the cascade. Loss of Hb results in absence of the first-born neuron identities in all neuroblast lineages assayed to date (1-1, 3-1, 3-5, 7-1, 7-3). Conversely, driving prolonged Hb expression in neuroblasts results in ectopic first-born neurons in all lineages tested (*Isshiki et al., 2001*; *Novotny et al., 2002*; *Kanai et al., 2005*; *Kohwi et al., 2013*). For example, prolonged expression of Hb in NB7-1 produces ectopic U1 motor neurons, whereas prolonged expression of Hb in NB7-3 produces ectopic EW1 serotonergic interneurons. Note that these misexpression experiments further confirm the neuroblast-specific effect of Hb, showing that the spatial identity of the neuroblast determines the effect of Hb. Importantly, Hb can induce early-born neuronal identity throughout a 'competence window' of ~5 neuroblast divisions (from embryonic stage 9–12). The length of the competence window is defined by expression of Distal antenna (Dan), a nuclear Pipsqueak domain protein present in all neuroblast nuclei until stage 12 (about five divisions for most neuroblasts); Dan is downregulated in all neuroblasts at the end of stage 12, and this closes the Hb competence window (*Kohwi et al., 2013*). Hb can induce first-born neuronal identity at any point during this competence window, showing that Hb binding sites are accessible throughout the competence window; this is important to consider for the experiments described here, where we have restricted our Hb binding and chromatin accessibility profiling experiments to the stage 9–12 competence window in individual neuroblast lineages (see below).

It is clear that spatial and temporal cues are integrated to generate lineage-specific neuronal diversity, both in Drosophila embryonic neuroblasts and optic lobe neuroblasts (*Erclik et al., 2017*), and likely in mammalian progenitor lineages. Yet in no case, mammals or Drosophila, is it known how spatial and TTFs are integrated. Here we hypothesise two mechanisms by which this integration could occur. (1) Independent specification (*Figure 1B*). In this scenario, spatial and temporal transcription factors bind their genomic targets independently, and the combinatorial actions of these factors and their downstream gene regulatory networks results in unique gene expression and therefore unique neural identities. (2) Sequential specification (*Figure 1C*). In this scenario, early expression of STFs in the neuroectoderm (where they are known to act) biases the subsequent DNA-binding profile of the later expressed TTFs. This could happen via STFs generating different chromatin landscapes in each neuroblast, or via STFs promoting the persistent expression of TTF cofactors that result in neuroblast-specific TTF DNA-binding. While both scenarios would result in the specification of distinct neural identities in spatially distinct NBs, in the independent specification model, TTF binding will be identical in all neuroblasts whereas in the sequential specification model, TTF binding will occur at different loci in each neuroblast.

To discriminate between these models, we sought to determine Hb genomic targets in NB5-6 versus NB7-4. If independent specification is used, we expect to find similar Hb occupancy in each neuroblast (*Figure 1B*), whereas if sequential specification is used, we expect to find different Hb genomic binding in each neuroblast (*Figure 1C*). Our goal was to identify Hb occupancy within the early NB5-6 and NB7-4 lineages during the Hb competence window, when Hb retains the ability to generate ectopic early-born neuronal identities, and thus presumably can still bind its normal genomic targets. To identify Hb occupancy in these two neuroblast lineages, we adapted the previously described Targeted DamID (TaDa) method (*Southall et al., 2013*; *Marshall et al., 2016*). TaDa relies

on an attenuated expression of the DNA adenosine methyltransferase (Dam) enzyme (*Figure 1D*), which binds genomic DNA and methylates adenosine at GATC sites. This covalent DNA mark can be used to determine Dam binding sites, due to the very low level of endogenous DNA methylation in Drosophila. Expression of Dam alone can be used to detect open chromatin (*Aughey et al., 2018*) (*Figure 1E*) or Dam can be fused to a transcription factor such as Hb, which provides a read-out of Hb genomic occupancy (*Figure 1F*).

Here we characterize two Gal4 lines that are specific for NB5-6 and NB7-4 lineages in the embryo. We use these lines to obtain NB-specific expression of Dam:Hb (to identify Hb genomic occupancy) and Dam alone (to detect open chromatin). We demonstrate that Hb has differential targets in NB5-6 and NB7-4 lineages, which correspond to differentially open chromatin in each lineage. Importantly, our observation that Hb-bound loci specific to NB5-6 have open chromatin, but the same loci in NB7-4 have closed chromatin, shows that Hb is not sufficient to create open chromatin. Rather, Hb binding in each neuroblast is likely restricted to a subset of neuroblast-specific open chromatin domains. In support of this model, the Gsb STF, required to specify NB5-6 but not NB7-4, shows enriched occupancy at open chromatin and Hb enriched loci in NB5-6, but not in NB7-4, consistent with a role for Gsb in generating neuroblast-specific open chromatin organization. Our findings support a sequential specification model in which STFs create neuroblast-specific chromatin organization, leading to neuroblast-specific Hb DNA-binding.

## Results

### Characterization of Gal4 lines specific for NB5-6 or NB7-4

Here we characterize two Gal4 lines that label either the NB5-6 or the NB7-4 lineages, which is a prerequisite for profiling neuroblast-specific Hb binding sites. NB5-6 forms in the Gsb domain, whereas NB7-4 forms in the Engrailed domain (*Figure 2A*). To label NB5-6 and its lineage we used *ladybird early (lbe)-Gal4*, which is reported to specifically label NB5-6 and its progeny (*Urbach and Technau, 2003*; *Baumgardt et al., 2009*). We confirmed that *lbe-Gal4* expression was highly specific to the NB5-6 and its lineage from stage 10 through stage 12, the time frame of our experiments (*Figure 2B–D'*; *Figure 2—figure supplement 1A*), although by stage 17 it has expression in the non-neuronal salivary gland (*Figure 2—figure supplement 1A*). Henceforth we call this line '*NB5-6-Gal4*.' To label NB7-4 and its lineage, we used the previously described *R19B03[AD] R18F07[DBD]* split-Gal4 line (*Lacin and Truman, 2016*). We confirmed that this line labels NB7-4 and its lineage from stage 10 until the end of stage 17 (*Figure 2E–G'*; *Figure 2—figure supplement 1B*); the only off-target expression is in the adjacent NB5-6 lineage in 6% of hemisegments (n = 1176). Henceforth we call this line '*NB7-4-Gal4*.' Both *NB5-6-Gal4* and *NB7-4-Gal4* lines are first expressed after Hb expression in the NB, but during the 'Hb competence window' defined by the presence of Distal antenna (Dan) nuclear protein in stage 9–12 neuroblasts (*Figure 2C' and F'*) (*Kohwi et al., 2013*). Importantly, ectopic Hb can induce early-born neuronal identity throughout the Hb competence window, and thus the relevant Hb DNA-binding sites are still accessible. We conclude that *NB5-6-Gal4* and *NB7-4-Gal4* lines are each expressed in a single neuroblast and its progeny during the Hb competence window and thus are ideal tools for expressing Dam or Dam:Hb in specific neuroblast lineages.

We next identified the early-born Hb+ progeny from both lineages, to ensure that each neuroblast lineage makes different Hb+ progeny. DiI clonal analyses show that both NB5-6 and NB7-4 make distinct populations of interneurons, but also similar populations of subperineurial glia, and their birth-order in the lineage has not been determined (*Schmidt et al., 1997*; *Schmid et al., 1999*). Therefore, we used *NB5-6-Gal4* to generate MultiColorFlipOut (MCFO; *Nern et al., 2015*) single neuron labelling among NB5-6 progeny. We repeatedly (n = 31) identified a Hb+ neuron that had a characteristic ipsilateral ascending projection, which we name the Chaise Lounge neuron due to its distinctive morphology; two segmentally repeated Chaise Lounge neurons are shown in *Figure 2H*; inset shows a Chaise Lounge neuron expressing Hb. We searched the EM reconstruction (*Ohyama et al., 2015*) and identified an identical Chaise Lounge neuron (*Figure 2I*). Thus, NB5-6 makes a distinctive ipsilateral neuron during its Hb expression window. Similarly, we used *NB7-4-Gal4* to generate MCFO single cell labelling, but could not directly identify a Hb+ neuron either due to loss of Hb from early-born neurons prior to neuronal differentiation, or due to lack of Gal4

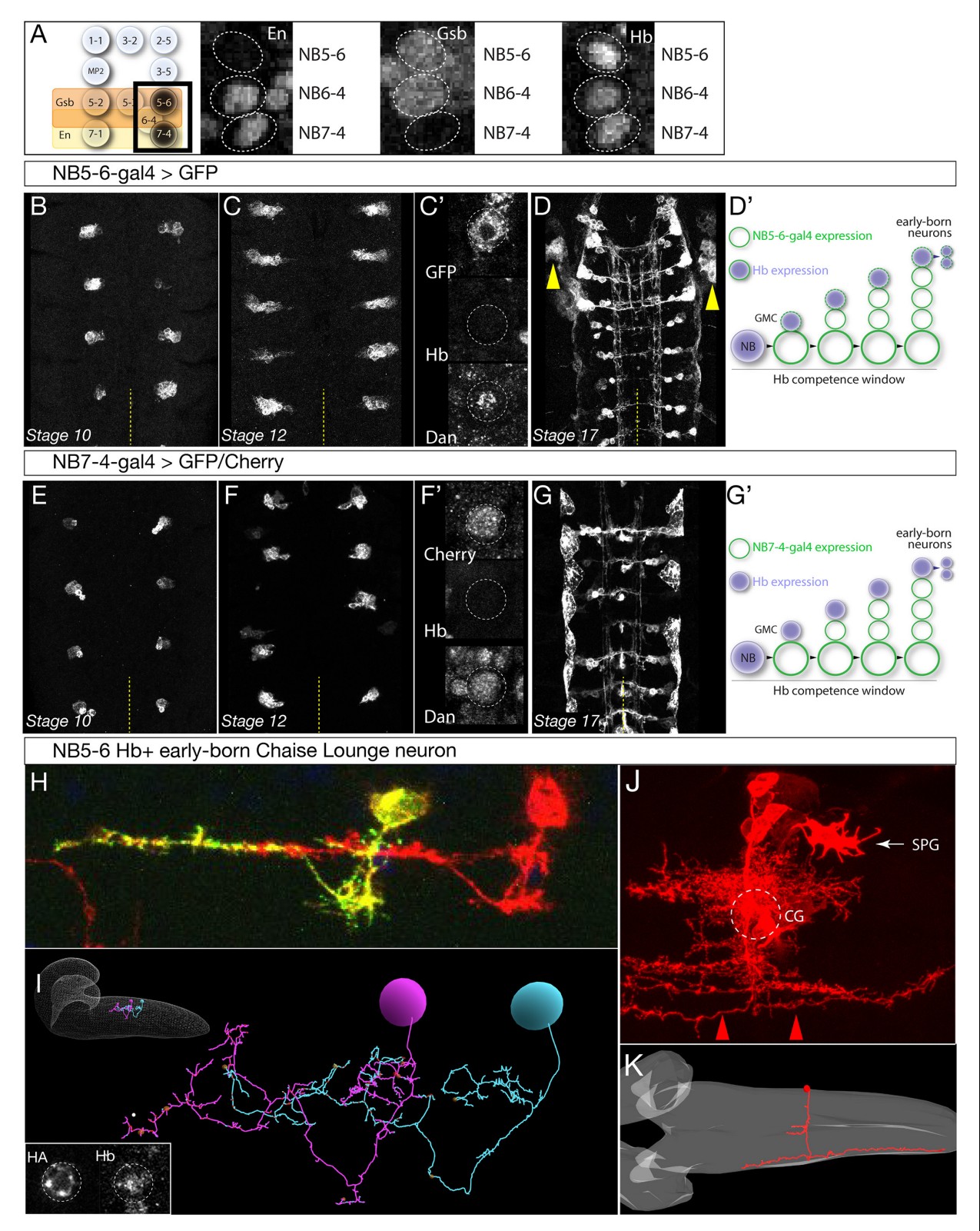

**Figure 2.** Identification of Gal4 lines specifically expressed in NB5-6 or NB7-4. (**A**) Left: schematic showing spatial positions of NB5-6 and NB7-4. Right: Immunostaining of stage nine embryos showing neuroblast-specific STF expression (En, Gsb) and common TTF expression (Hb). Genotype: *en-Gal4/ UAS-GFP*. (**B–D'**) *NB5-6-Gal4* is expressed in the NB5-6 lineage from stage 10 until the end of embryogenesis. Dan is present in NB5-6 through stage 12 (**C'**). (**D'**) Schematic of NB5-6 expression (green outlines) and Hb expression (purple), see text for details. Note that Gal4 expression is present during

*Figure 2 continued on next page*

Figure 2 continued

the Dan + Hb competence window. Genotype: *lbe-K-Gal4/UAS-myr::GFP*. (E–G') *NB7-4-Gal4* is expressed in the NB7-4 lineage from stage 10 until the end of embryogenesis. Dan is present in NB5-6 through stage 12 (F'). (G') Schematic of NB7-4 expression (green outlines) and Hb expression (purple), see text for details. Genotype: *R19B03^{AD}/UAS-myrGFP; R18F07^{DBD}/+*. (H–I) NB5-6 early-born Chaise Lounge neurons. Lateral view, anterior, left. (H) Two segmentally repeated Chaise Lounge neurons labelled by MCFO (*hs-FLP lbe-K-Gal4 UAS-MCFO*); the Chaise Lounge neurons are Hb+ (inset). Note the ipsilateral projections. (I) Two segmentally repeated Chaise Lounge neurons in the EM reconstruction, where they are named A27k. Inset: outline of CNS with Chaise Lounge neurons shown. (J–K) NB7-4 early-born G neuron. (J) MARCM clone made with *en-Gal4* labels most or all of the NB7-4 lineage, including the diagnostic Channel Glia (CG) which are only made by NB7-4 (*Schmidt et al., 1997; Schmid et al., 1999*). Note the G neuron axon arbors which project the most laterally in the connective and are both ascending and descending (red arrowheads). SPG, subperineurial glia. Dorsal view, anterior to left. (J) The G neuron in the EM reconstruction (red). The neuropil is outlined in gray. Note the lateral axon projection that is ascending and descending, and the cell body position contacting the neuropil. Also note the two small bilateral midline processes, which match those of the grasshopper G neuron (*Raper et al., 1983*).

DOI: https://doi.org/10.7554/eLife.44036.004

The following figure supplement is available for figure 2:

**Figure supplement 1.** Expression pattern of NB5-6 and NB7-4 Gal4 lines.

DOI: https://doi.org/10.7554/eLife.44036.005

expression in these neurons. Instead, we used multiple criteria to identify a putative early-born neuron, the G neuron, using MARCM clones (*Figure 2J*), and EM reconstruction (*Figure 2K*). Our criteria for assigning this neuron as early-born include (i) presence of the neuron in full NB7-4 clones (*Figure 2J*) but not in the NB7-4-Gal4 pattern (*Figure 2—figure supplement 1*), which misses early-born neurons; (ii) cell body position next to the neuropil, where most Hb+ neurons are located (*Kambadur et al., 1998*); and (iii) close morphological match to the grasshopper G neuron, an early-born neuron from NB7-4, including ascending and descending projections in the most lateral connective tract (*Raper et al., 1983*). Finally, we note that all NB7-4 neuronal progeny have contralateral axons (*Schmidt et al., 1997; Schmid et al., 1999*), whereas the NB5-6 early-born Chaise Lounge neuron has ipsilateral projections. Thus, we conclude that NB5-6 and NB7-4 produce different neurons during the Hb expression window. This makes NB5-6 and NB7-4 an appropriate model system to characterize how different spatial patterning cues produce distinct Hb+ early born cell types.

## Generation of a functional, non-toxic Dam:Hb fusion protein

The first step in using the TaDa method to map Hb occupancy in the NB5-6 and NB7-4 lineages is to generate a functional, non-toxic Dam:Hb fusion protein. Although other Dam constructs have been shown to be non-toxic (*Southall et al., 2013; Marshall et al., 2016; Aughey et al., 2018*), this is the first use of Dam:Hb and its toxicity is unknown. We used standard methods to generate a *UAS-LT3-Dam:hb* transgene where the first open reading frame (ORF) encodes Cherry and the second ORF encodes Dam:Hb (see *Figure 1D,F*); placing the Dam fusion protein in the second ORF is important to keep both Dam and Hb levels extremely low, which reduces toxicity and increases specificity of DNA binding (*Southall et al., 2013*).

To determine if Dam:Hb is toxic, we expressed the fusion protein throughout the nervous system (*sca-Gal4 UAS-Dam:Hb*) and ubiquitously (*Da-Gal4 UAS-Dam:Hb*), and observed no effect on embryonic viability (*Figure 3A*). To determine whether the Hb portion of the Dam:Hb fusion protein was functional, we assayed for its ability to generate ectopic Eve+ U neurons, despite being expressed at very low levels. In wild type, NB7-1 generates five Eve+ U neurons, including the Hb+ early born U1 and U2 neurons, and extending neuroblast expression of Hb produces many ectopic Eve+ U1/U2 neurons (*Isshiki et al., 2001; Pearson and Doe, 2003*). We observed that expression of Dam:Hb was capable of inducing a small number of ectopic Eve+ neurons (*Figure 3B*), despite the low levels of Dam:Hb, showing that Dam:Hb is functional. We conclude that Dam:Hb is non-toxic in embryos, and that it is functional for inducing early-born neuronal identity.

The fact that Dam:Hb can induce early-born neuronal identity suggests that it can bind the same genomic targets as Hb, but we wanted to determine this important point experimentally. The TaDa method involves comparing Dam genomic binding to Dam:Hb genomic binding, with a normalised ratio used to identify sites preferentially bound by the Dam:Hb fusion protein (*Southall et al., 2013; Marshall and Brand, 2015*). We expressed Dam or Dam:Hb in all cells throughout embryogenesis,

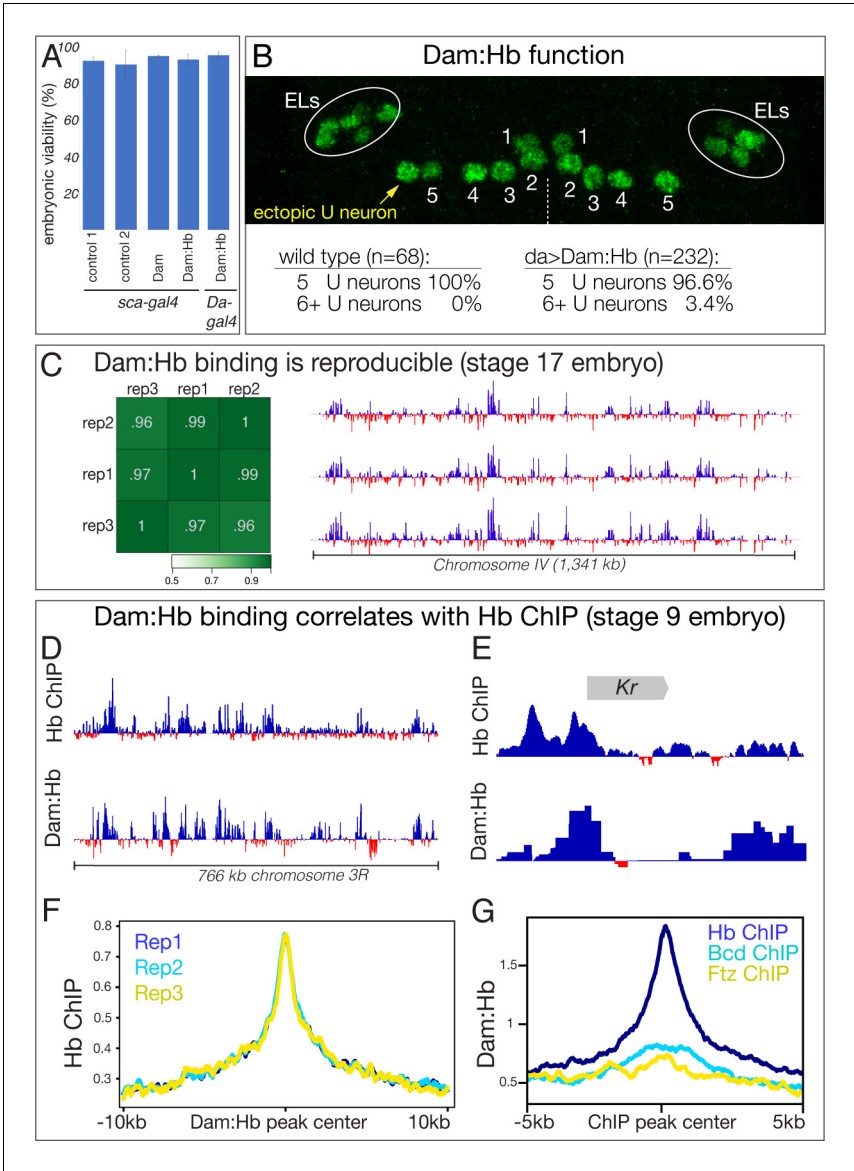

**Figure 3.** Generation of a functional, non-toxic Dam:Hb fusion protein. (**A**) Low level Dam:Hb expression is non-toxic. Control 1, *sca-gal4/sca-gal4*; control 2, *sca-gal4 UAS-HA::UPRT* (*Miller et al., 2009*); Dam, *sca-gal4 UAS-LT3-Dam*; Dam:Hb, *sca-gal4 UAS-LT3-Dam:Hb*, Dam:Hb, *Da-gal4 UAS-LT3-Dam:Hb* (n = 300 for each genotype). (**B**) Dam:Hb retains Hb function and can induce ectopic Eve+ U neurons. Anterior up; midline, dashed line. Left hemisegment shows a single ectopic Eve+ neuron (yellow) to comprise six total U neurons, whereas the right hemisegment has the normal five U neurons. Below, quantification. Wild type (*y w*) represents 68 hemisegments from six embryos; Dam:Hb (*da-Gal4 UAS-LT3-Dam:Hb*, second ORF) represents 8 of 232 hemisegments from 15 embryos with an ectopic U neuron. ELs, Eve lateral neurons. (**C**) Dam:Hb binding is reproducible. Left, three biological replicates of genomic binding sites showing high Pearson correlation coefficients. Right, Dam:Hb binding over 1341 kb on chromosome IV is highly similar in all three biological replicates. Genotype *da-Gal4 UAS-LT3-Dam:Hb* in stage 17 embryos. Data range: −2.84–7.07. (**D–G**) Dam:Hb-bound loci correlate with Hb ChIP loci. (**D**) Alignment of Dam:Hb and Hb ChIP binding sites over 766 kb of genomic DNA near the Hb locus, where Hb is known to bind. Data range for Hb ChIP: −1.01–6.23; Data range for Dam:Hb: −2.63–5.3. (**E**) Alignment of Dam:Hb and Hb ChIP binding sites at the *Krüppel* (*Kr*) locus. Data range for Hb ChIP: −1.66–9.04; Data range for Dam:Hb: −0.63–5.68. (**F**) Dam:Hb peaks for three replicates (blue, cyan, yellow) are correlated with Hb ChIP signal. Plot shows the Hb ChIP signal ±10 kb of the center of all the peaks identified by Dam:Hb analysis in the three replicates. (**G**) Dam:Hb signal is enriched at sites of Hb ChIP binding (blue), but not that of Bcd (cyan) or Ftz (yellow). Plot shows the Dam:Hb signal ±5 kb of the center of all the peaks identified by ChIP-chip analysis.
DOI: https://doi.org/10.7554/eLife.44036.006

*Figure 3 continued on next page*

*Figure 3 continued*

The following figure supplements are available for figure 3:

**Figure supplement 1.** Dam:Hb and Hb-ChIP show similar binding at known Hb target genes.

DOI: https://doi.org/10.7554/eLife.44036.007

**Figure supplement 2.** Dam:Hb and Hb-ChIP binding is correlated.

DOI: https://doi.org/10.7554/eLife.44036.008

**Figure supplement 3.** Hb binding motifs are enriched at Dam:Hb bound loci.

DOI: https://doi.org/10.7554/eLife.44036.009

measured the quantile normalised ratio between them to identify Dam:Hb binding sites (see Materials and methods), and performed three biological replicates at embryonic stage 17. We found that the biological replicates showed high Pearson correlation coefficients (*Figure 3C*, left), and were qualitatively very similar along the entire fourth chromosome (*Figure 3C*, right). Most importantly, we compared Dam:Hb genomic occupancy with published Hb genomic occupancy determined by chromatin immunoprecipitation (ChIP) (*Li et al., 2008*; *Bradley et al., 2010*). A comparison over 700 kb of genomic DNA on chromosome 3R showed qualitatively similar Dam:Hb and Hb ChIP binding profiles (*Figure 3D*). Indeed, enriched Dam:Hb binding was detected at eight of the nine known Hb target genes (*Lyne et al., 2007*) (*Figure 3E*, *Figure 3—figure supplement 1*). We next compared the similarities in Hb occupancy as reported by these two techniques at the genomic level. To do this, we ran the MACS2 peak caller (*Zhang et al., 2008*) on the two datasets and identified 6597 and 6656 regions significantly enriched for Dam:Hb and Hb ChIP respectively (see Materials and methods). We found that 1972 regions were shared between the two (29.89% of ChIP peaks and 29.62% of Dam:Hb peaks). When broad peaks were used for this analysis, 2394 regions were shared between the two, or 33.74% of ChIP peaks and 45.13% of Dam:Hb peaks; and when the narrow peaks were extended to 2 kb on either side of the peak summit, 2207 regions were shared between the two, or 57.53% of ChIP peaks and 60.37% of Dam:Hb peaks. A Monte Carlo analysis on the narrow peak overlap showed this was highly significant, detecting only 6.16% overlap with a set of random peaks (100 iterations, p-value $< 1\ e^{-300}$, see Materials and methods). Correspondingly, we found high ChIP signals at the Dam:Hb binding sites and vice versa (*Figure 3F, G*, *Figure 3—figure supplement 2*). Importantly, this overlap in occupancy was not seen when the Dam:Hb data were compared with the ChIP-seq data of any other transcription factor, such as Ftz or Bcd (*Figure 3G*), demonstrating the specificity of the method. Additional support for the accuracy of Dam:Hb binding is that the known Hb DNA-binding motif is the most enriched motif at Dam:Hb binding sites (*Figure 3—figure supplement 3*). Taken together, these results show that Dam:Hb binding closely mimics endogenous Hb binding.

## NB5-6 and NB7-4 lineages have different Hb-bound loci

At this point we have validated two neuroblast-specific Gal4 lines, as well as shown that Dam:Hb genomic binding is both reproducible and matches published Hb ChIP data in stage nine whole embryos. However, to test the two models of spatial and temporal integration we had to use Dam:Hb in the NB5-6 or NB7-4 lineages – much smaller pools of cells – to determine whether Hb genomic targets were the same or different in these spatially distinct NB lineages. Therefore, our next step was to determine if we could get reproducible Dam:Hb binding data from this small pool of cells, and with shorter Dam:Hb exposure than previously reported (*Southall et al., 2013*; *Erclik et al., 2017*; *Widmer et al., 2018*). For this purpose, we modified the published protocol to allow processing of more starting material (see Materials and methods). We expressed Dam:Hb in a single neuroblast lineage in each hemisegment (about 200 cells in the ~50,000 cell embryo) and for five hours (from embryonic stage 9–12). Previous experiments had expressed Dam constructs in a higher fraction of cells and for $\geq$12 hr (*Southall et al., 2013*; *Cheetham et al., 2018*; *Widmer et al., 2018*). We expressed Dam:Hb using each of two neuroblast-specific Gal4 lines (*NB5-6-Gal4* and *NB7-4-Gal4*) and purified DNA from stage 12 embryos, near the end of the Hb competence window (see Materials and methods). We performed three biological replicates for each neuroblast and observed excellent reproducibility across all replicates (*Figure 4A*). We conclude that we can get a reproducible Dam:Hb signal from a single neuroblast lineage during the Hb competence window.

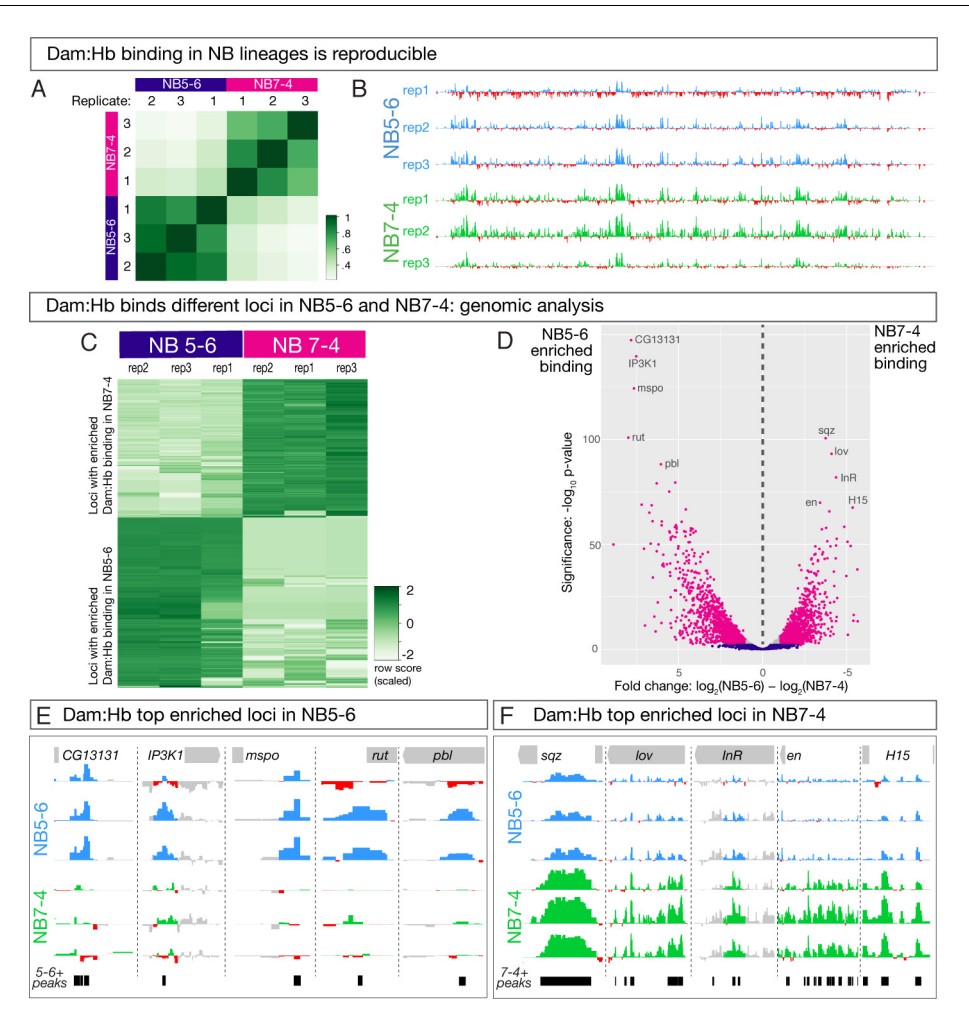

**Figure 4.** Dam:Hb has distinct genomic binding sites in NB5-6 and NB7-4 lineages. (**A,B**) Dam:Hb binding in the NB5-6 lineage and the NB7-4 lineage is reproducible. (**A**) Three biological replicates of Dam:Hb in each neuroblast lineage are shown, with high Pearson correlation coefficients within each neuroblast replicate, and low correlation coefficients between each neuroblast. (**B**) Dam:Hb binding over 1,341 kb on chromosome IV is qualitatively similar between lineages. Data range: −3.49–8.71. (**C–F**) Differential binding data showing Dam:Hb binds different loci in NB5-6 versus NB7-4. (**C**) A binding affinity heatmap (scaled) showing reads at loci differentially occupied by Dam:Hb in NB5-6 and NB7-4. Loci (rows) are shown for biological replicates of both neuroblasts with greater densities of Dam:Hb binding in darker colours. Note that sites with higher counts in the three NB7-4 replicates (top right) are depleted in the three NB5-6 replicates (top left), and vice versa. (**D**) Volcano plot showing differentially occupied loci that are FDR ≤ 0.01 in magenta, FDR > 0.01 in blue, and those that have a fold change of less than two in grey. This threshold corresponds to 718 loci in NB5-6 and 504 loci in NB7-4. Genome-wide Hb-bound loci in both neuroblasts were analysed for differential analysis using DiffBind (***Ross-Innes et al., 2012***) with DESeq2 and edgeR and two independent peakcallers with similar results. These plots show DESeq2 results with the MACS2 peak caller (***Zhang et al., 2008***). (**E,F**) The top five enriched Dam:Hb-bound loci are shown for NB5-6 (blue track in F) versus NB7-4 (green tracks in G) lineages. The black bars represent the loci identified as differentially bound in the analysis. Data range: −1.9–3.96. For all panels, NB5-6 genotype: *NB5-6-Gal4 UAS-LT3-Dam:Hb* or *UAS-LT3-Dam*. NB7-4 genotype: *NB7-4-Gal4 UAS-LT3-Dam:Hb* or *UAS-LT3-Dam*.
DOI: https://doi.org/10.7554/eLife.44036.010

The following figure supplement is available for figure 4:

**Figure supplement 1.** Dam:Hb shows similar binding at loci expressed in both NB5-6 and NB7-4.
DOI: https://doi.org/10.7554/eLife.44036.011

Next, we wanted to determine whether Dam:Hb binds the same or different loci in the two different neuroblasts. The high correlation between biological replicates for each neuroblast, plus the lack of correlation between the two neuroblasts, provided a gross indication that Dam:Hb has unique binding sites in each neuroblast lineage (*Figure 4A*). We expected the number of differentially bound loci to be relatively small, because most genes are not predicted to regulate NB5-6/NB7-4 differences, and indeed, comparing Hb binding along the entire fourth chromosome shows qualitative similarities between the two NB lineages (*Figure 4B*). This is also evident at genes known to be expressed in and regulated by Hb across many neuroblast lineages – for example *Kr*, *pdm2* and *zfh2* (*Isshiki et al., 2001*) (*Figure 4—figure supplement 1*). These similarities confirm the reproducibility of Dam:Hb binding in two distinct neuroblast lineages.

To begin our analysis of differential Dam:Hb binding between NB5-6 lineage and NB7-4 lineages, we first ran the MACS2 peak caller (*Zhang et al., 2008*) on the six datasets – three replicates of NB5-6 lineage and three replicates of NB7-4 lineage – to identify regions significantly bound by Hb in each sample. The rest of our analyses focussed on the significantly bound Hb loci in the two NB lineages. We used the *R* Bioconductor package DiffBind (*Ross-Innes et al., 2012*) to identify 4224 differentially bound loci in the two NB lineages: 2007 that were enriched for Dam:Hb binding in the NB5-6 lineage, and 2217 that were enriched for Dam:Hb binding in the NB7-4 lineage (*Figure 4C*; *Supplementary file 1*). In addition, there were 2860 loci occupied by Dam:Hb in both neuroblast lineages (*Supplementary file 1*). Importantly, while the read densities at individual loci are similar between replicates, they are strikingly different between the two neuroblast lineages.

Next we represented the differentially bound loci using a volcano plot, where the magenta dots highlight the most significantly differential loci with more than 2-fold change and an FDR of $\leq$0.01 (*Figure 4D*). This threshold corresponds to 718 Hb enriched loci in NB5-6 lineage and 504 Hb enriched loci in NB7-4 lineage (*Supplementary file 1*), which is what we use for all subsequent analyses. The genes closest to the top five differentially occupied loci in each neuroblast are marked in this plot, and shown in *Figure 4E,F*. Based on these results, we conclude that Dam:Hb binds different loci in different neuroblasts. This clearly rules out the independent specification model where Hb has identical binding sites in different neuroblasts.

## Different chromatin states in NB5-6 and NB7-4 lineages

We next wanted to understand how STFs might influence TTF genomic binding. Given the order of their action – STFs acting early in the neuroectoderm, and TTFs acting later in the delaminated NB – one possibility is that STFs generate different open/closed chromatin landscapes in each neuroblast such that TTFs have access to different loci in each neuroblast. This would predict that spatially distinct NBs would have different open/closed chromatin landscapes. To determine if this were indeed true, we performed chromatin accessibility profiling by Dam only (CaTaDa), which exploits the ability of the Dam protein to bind open chromatin domains (*Aughey et al., 2018*). We first expressed Dam in all cells throughout embryogenesis using *Da-Gal4* and observed excellent reproducibility between biological replicates both qualitatively and quantitatively (*Figure 5A*, red tracks in C). We next wanted to confirm that Dam only binding in the embryo correlates with open chromatin domains, as has been shown in other cell types (*Aughey et al., 2018*). To do this, we analysed the Dam only signal around the DNase I hypersensitive sites (peaks) made available by the BDTNP consortium (*Thomas et al., 2011*) and found enriched Dam signals around the DNaseI peaks, as well as qualitative similarities between the two (*Figure 5B*, compare red and ochre tracks in C). We observed 6,708 Dam only peaks were aligned with DNase I hypersensitive peaks (44.6% of all Dam only peaks; 33.9% of all DNaseI peaks). A Monte Carlo analysis showed this was highly significant, detecting only 18.14% overlap with a set of random peaks (100 iterations, p-value $< 1\ e^{-300}$, see Materials and methods). These data suggest that Dam only can be used to detect open chromatin in embryos.

We next sought to determine whether Dam only could be used to assay open chromatin in small pools of cells over a short period of time – for example in NB5-6 and NB7-4 lineages at stage 12. We performed three biological replicates of Dam only for each neuroblast, and observed excellent reproducibility in all but one replicate, so we used the two best replicates henceforth (*Figure 5D*). The reproducibility of the method can also be observed in the similar Dam binding patterns seen at representative control genes that are equally expressed in NB5-6 and NB7-4 lineages (e.g. *Kr*, *pdm2* and *zfh2*), or along a large stretch of chromosome 4 (*Figure 5—figure supplement 1*).

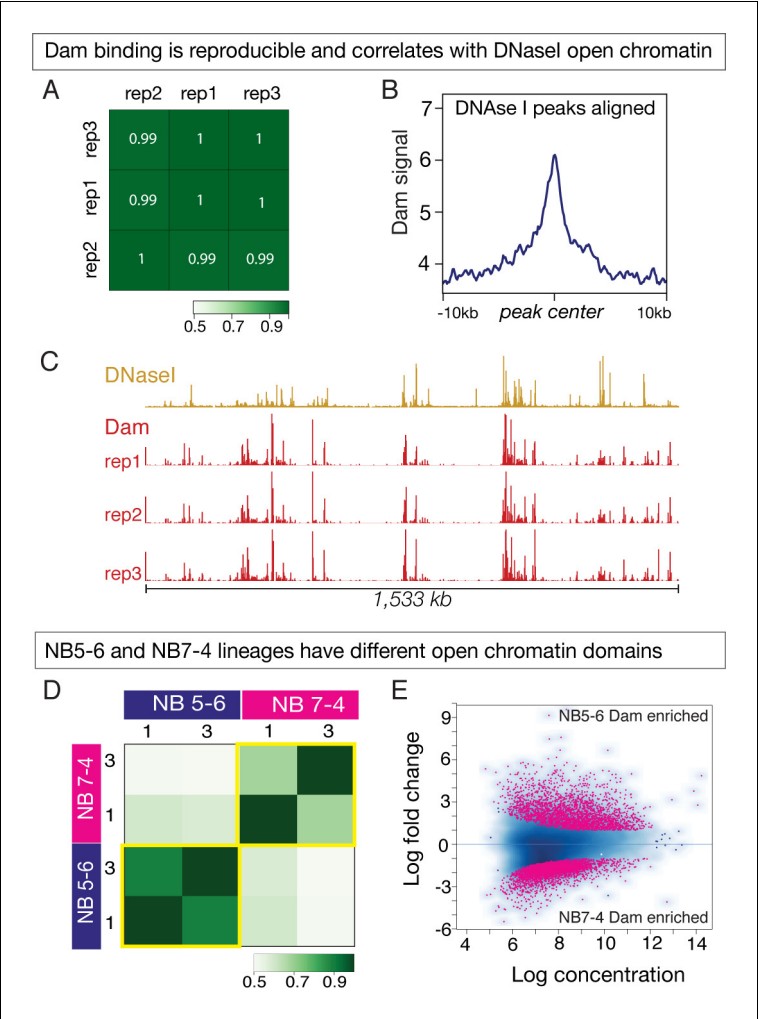

**Figure 5.** Dam only binding shows differential open chromatin landscapes in NB5-6 and NB7-4 lineages. (**A–C**) Dam binding is reproducible and correlates with DNAse I sites. (**A**) Three biological replicates are shown, with high Pearson correlation coefficients. (**B**) Dam binding is enriched at DNAse I hypersensitive peaks. (**C**) Dam binding over 1,533 kb on chromosome 3R is similar in all replicates (red tracks), and similar to DNAseI hypersensitivity data (ochre tracks). Data range for Dam: 0–50; Data range for DNAseI: 0–150. Genotype: *Da-Gal4/ UAS-LT3-Dam*. (**D–E**) Dam binding reveals different open chromatin domains in NB5-6 versus NB7-4. (**D**) Heat map showing Dam binding sites in NB5-6 have high Pearson correlation coefficients in two replicates, but note the low correlation coefficients between NB5-6 and NB7-4 replicates, showing that each neuroblast has different open chromatin landscapes. (**E**) Dam binds different loci in the NB5-6 lineage versus the NB7-4 lineage. MA plot showing 3656 loci enriched for Dam binding in the NB5-6 lineage (top) and 5084 loci enriched for Dam binding in the NB7-4 lineage (bottom).

DOI: https://doi.org/10.7554/eLife.44036.012

The following figure supplement is available for figure 5:

**Figure supplement 1.** Dam only shows similar binding at loci expressed in both NB5-6 and NB7-4.
DOI: https://doi.org/10.7554/eLife.44036.013

Next, we investigated whether there were global differences in chromatin states between the two neuroblast lineages. To do this, we first determined regions of significantly open chromatin in the two neuroblast lineages by running the MACS2 peak caller (*Zhang et al., 2008*) on the four best replicates, which gave us a 'peakset' of significantly open chromatin in NB5-6 and NB7-4 lineages. We used these regions of open chromatin in both NB5-6 and NB7-4 lineages to conduct a differential analysis using the DiffBind package (*Ross-Innes et al., 2012*) and identified a total of 8,740 Dam only differentially bound loci, including 3656 loci in the NB5-6 lineage and 5084 loci in the NB7-4

lineage. These regions of differential chromatin accessibility have been represented as an 'MA plot' with the NB5-6 differential open chromatin loci at the top and the NB7-4 differential open chromatin loci at the bottom (*Figure 5E*). We conclude that there are global differences in the open chromatin landscape between the NB5-6 and NB7-4 lineages.

## Neuroblast-specific Hb-bound loci correlate with neuroblast-specific open chromatin domains

Chromatin accessibility has been shown to be the strongest determinant of TF occupancy on the genome (*Li et al., 2008*; *Kaplan et al., 2011*; *Guertin et al., 2012*). We wanted to determine if Dam:Hb binding was similarly responsive to the state of the chromatin in the NB5-6 and NB7-4 lineages. To do this, we took all Dam:Hb-bound loci – both those specific for each neuroblast as well as those shared by both neuroblasts – and queried the state of the chromatin at these loci in each NB lineage. We found that Dam:Hb-bound loci in the NB5-6 lineage were enriched for open chromatin in that lineage (*Figure 6—figure supplement 1A*), and similarly, Dam:Hb-bound loci in the NB7-4 lineage were enriched for open chromatin in that lineage (*Figure 6—figure supplement 1B*). This suggests that Dam:Hb binding is indeed correlated with chromatin accessibility domains in both NB lineages (*Figure 6—figure supplement 1C*).

If Dam:Hb preferentially occupies regions of open chromatin, we reasoned that the differentially occupied Dam:Hb loci in each NB lineage (lineage-specific Hb loci) must be correlated with differentially open chromatin in that neuroblast lineage (lineage-specific open chromatin). Indeed, NB5-6-specific Dam:Hb bound loci showed a strong enrichment for open chromatin (*Figure 6A*, blue lines); strikingly, these same loci had closed chromatin in NB7-4 (*Figure 6A*, green lines). Similarly, NB7-4-specific Dam:Hb bound loci showed strong enrichment for open chromatin (*Figure 6B*, green lines), while these same loci had closed chromatin in NB5-6 lineage (*Figure 6B*, blue lines). Corresponding to this, we found 364 peaks, or 50.76% of the differential Dam:Hb peaks in NB5-6 overlapped with differentially open chromatin peaks in that lineage; and 164 peaks or 32.74% of the differential Dam: Hb peaks in NB7-4 overlapped with differentially open chromatin peaks in that lineage. A Monte Carlo analysis showed these overlaps to be highly significant, detecting 5.23% overlap with a set of random peaks in NB5-6% and 6.75% in NB 7–4 (100 iterations, p-value $< 1 \, e^{-300}$ for NB 5–6 and $8.9 \, e^{-133}$ for NB 7–4, see Materials and methods). As a control, we assayed loci bound by Dam:Hb in both neuroblast lineages and found that there was no difference between lineages in open chromatin at these sites (*Figure 6C*). We confirmed these findings at the top five differentially bound Dam: Hb loci in the two neuroblast lineages. All but two of these differentially bound loci were also identified in the differential chromatin analysis; even the two that were not picked up in the analysis (*sqz* and *mspo*) were qualitatively different between the two neuroblast lineages (*Figure 6D,E*). We conclude that neuroblast-specific Dam:Hb binding occurs within neuroblast-specific accessible chromatin domains. This correlation suggests that either Hb binds where chromatin is open, or that Hb binding opens chromatin. The latter model seems unlikely, because both NB5-6 and NB7-4 are exposed to Hb expression, yet each neuroblast has specific open chromatin domains (see Discussion). We favor a model in which STFs generate neuroblast-specific open chromatin domains, leading to neuroblast-specific Hb occupancy.

## The row five spatial transcription factor gsb is enriched at open chromatin and Hb-bound loci in NB5-6, but not NB7-4

If spatial factors generate lineage-specific chromatin landscapes as the sequential specification model proposes, then it's likely that lineage-specific STF occupancy will correspond to lineage specific chromatin accessibility. Gsb is one of the best studied STFs in the embryonic VNC. It has been shown to be both necessary and sufficient to determine the identity of the row 5 NBs (*Skeath et al., 1995*; *Bhat, 1996*). Not only is Gsb a functionally validated STF, but Gsb ChIP-chip data from 0 to 12 hr embryos are publicly available (*Bonneaud et al., 2017*). As NB5-6 is a row 5 NB lineage specified by Gsb, it gave us the opportunity to test the sequential specification model more deeply. We asked whether Gsb occupancy was enriched at regions of accessible chromatin in the NB5-6 lineage. We plotted the Gsb ChIP-chip signal around all NB5-6 open chromatin loci and compared this with Gsb ChIP-chip signal around NB7-4 open chromatin loci. Indeed, we found an enrichment of Gsb signal specifically around NB5-6 open chromatin and not NB7-4 open chromatin (*Figure 7A*). A

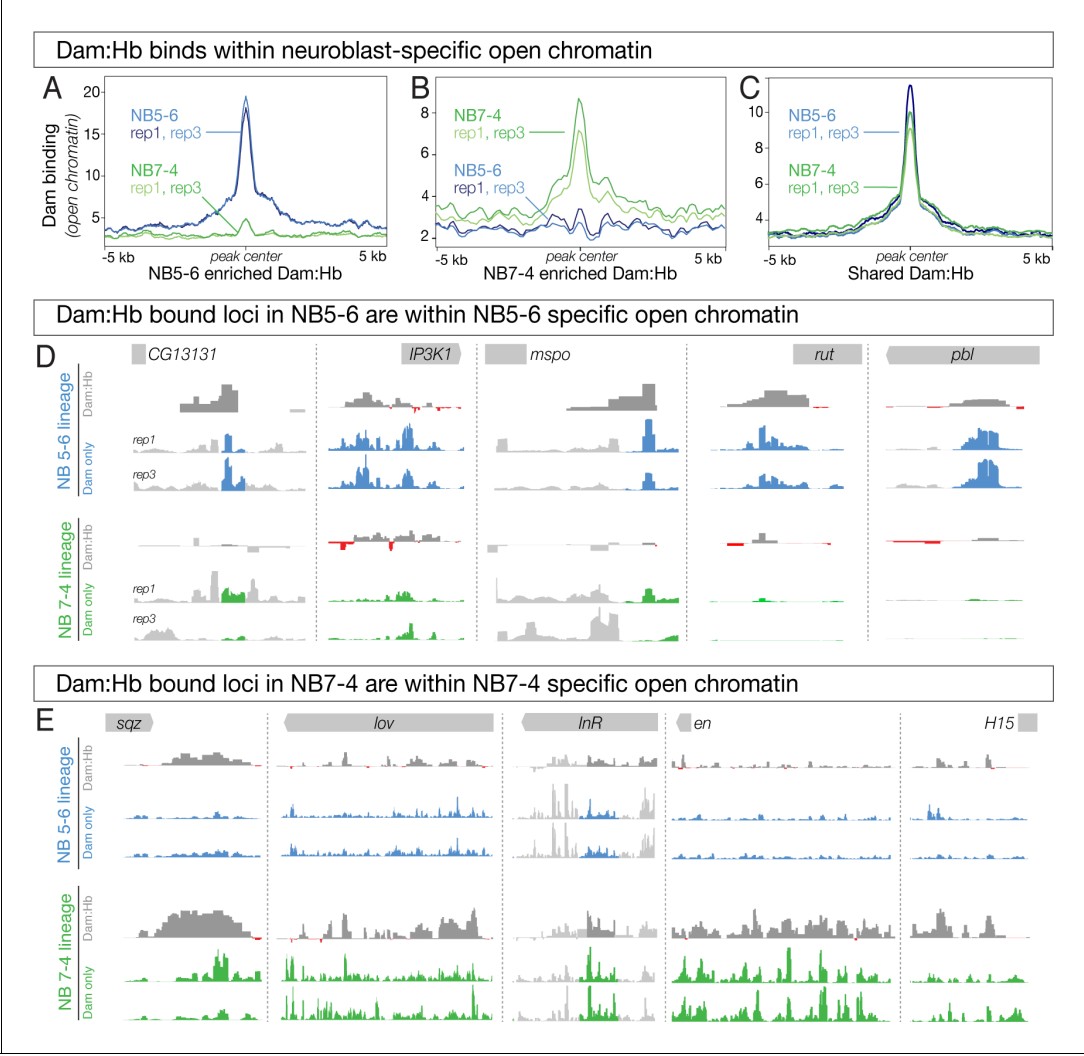

**Figure 6.** Differential chromatin in the 5–6 and 7–4 neuroblast lineages is correlated with differential Hb occupancy. (**A–C**) Dam:Hb binds within neuroblast-specific open chromatin. (**A**) Dam signal (open chromatin) in NB 5–6 (blue lines) and NB 7–4 (green lines) at loci where Dam:Hb binding is enriched in NB5-6 over NB7-4. Note that the chromatin is more open in NB5-6 than in NB7-4 at these loci. (**B**) Dam signal (open chromatin) in NB 7–4 (green lines) and NB 5–6 (blue lines) at loci where Dam:Hb binding is enriched in NB7-4 over NB5-6. Note that the chromatin is more open in NB7-4 than in NB5-6 at these loci. (**C**) Dam signal (open chromatin) at loci similarly occupied by Hb in both NB5-6 and NB7-4 lineages. (**D**) The top five Dam:Hb enriched loci in NB5-6 are in regions of NB5-6 open chromatin (blue tracks); however, in NB7-4 these loci are not in open chromatin (Dam; green tracks), and are not bound by Dam:Hb. Rows from top to bottom: genomic locus, Dam:Hb enrichment in NB5-6, Dam only enrichment in two replicates in NB5-6, Dam:Hb enrichment in NB7-4, and Dam only enrichment in two replicates in NB7-4. Data range for *IP3K1*, *rut*, *pbl* is 0–109; data range for *CG13131* and *mspo* is 0–15. (**E**) The top five Dam:Hb enriched loci in NB7-4 are in regions of open chromatin in NB7-4 (green tracks); however, in NB5-6 these loci are not in open chromatin (Dam; blue tracks) and are not bound by Dam:Hb. Rows from top to bottom: genomic locus, Dam:Hb enrichment in NB5-6, Dam only enrichment in two replicates in NB5-6, Dam:Hb enrichment in NB7-4, and Dam only enrichment in two replicates in NB7-4. Data range for *sqz*, *InR* and *en* is 0–35; data range for *lov* and *H15* is 0–20.
DOI: https://doi.org/10.7554/eLife.44036.014

The following figure supplement is available for figure 6:

**Figure supplement 1.** Dam:Hb binding is biased towards regions of open chromatin.
DOI: https://doi.org/10.7554/eLife.44036.015

Monte Carlo analysis found this enrichment to be highly significant (average real NB5−6/NB7-4 fold change = 2.198, average simulated NB5−6/NB7-4 fold change = 0.922, 100 random iterations, p-value = $1.19119 \, e^{-62}$). This supports the hypothesis that lineage-specific STFs generate lineage-specific chromatin landscapes.

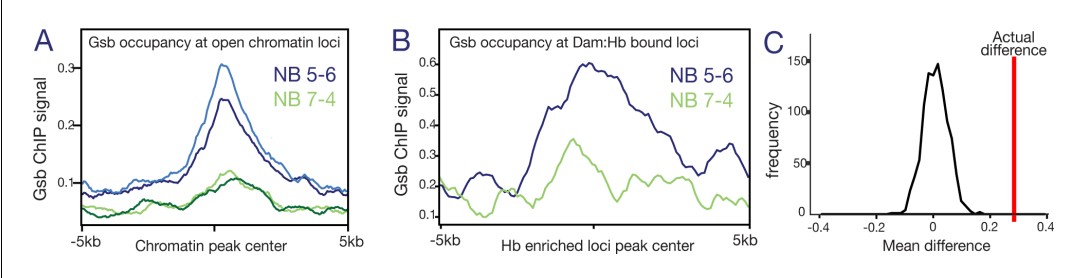

**Figure 7.** Gsb binding is enriched at open chromatin and Dam:Hb bound loci in NB5-6, but not NB7-4. (**A**) Gsb ChIP-chip signal at the regions of Dam-bound (open) chromatin; note the enrichment in NB5-6 (blue lines) but not NB7-4 (green lines). The number of peaks used is 20,838 and 18,201 for the NB5-6 reps and 29,817 and 31,080 for NB7-4. (**B**) Gsb ChIP-chip signal at the regions of Dam:Hb bound loci; note the enrichment in NB5-6 (blue lines) compared to NB7-4 (green lines). The number of peaks used is 504 and 718 in the two NBs, respectively. (**C**) Monte Carlo analysis shows that the average enrichment of Gsb signal around the actual NB5-6 loci (red line) is significantly higher than the distribution of average signal calculated for a similar number of random loci (1000 iterations, black line).

DOI: https://doi.org/10.7554/eLife.44036.016

Finally, we reasoned that if Hb preferentially binds to regions of accessible chromatin, and STF occupancy correlates with open chromatin in a lineage-specific manner, then the lineage-specific Hb occupancy that we observe in NB5-6 should correlate with lineage specific STF occupancy. We therefore plotted Gsb signal around NB5-6-enriched Hb loci and found a corresponding enrichment of Gsb occupancy at these regions (*Figure 7B*, blue line). In contrast, the NB7-4-enriched Hb loci did not show any such enrichment (*Figure 7B*, green line). A Monte Carlo analysis found this enrichment to be highly significant (average real NB5-6/NB7-4 fold change = 2.2, average simulated NB5-6/NB7-4 fold change = 1.2, 1000 random iterations, p-value = 6.54 e$^{-10}$; see Materials and methods). *Figure 7C* represents this analysis graphically: the real signal difference between NB5-6 and NB7-4 (*Figure 7C*, red line) is much greater than the distribution of differences calculated over the 1000 random iterations (*Figure 7C*, black line). Furthermore, we found that of the 503 Hb enriched loci in NB5-6, 101 had a Gsb peak within 2 Kb of the centre, whereas this number was 49 for NB7-4. A Fisher's exact test on these data found this spatial relationship to be highly significant for NB5-6 (p = 8.78e-19), but not for NB7-4 (p = 0.078). We conclude that loci differentially bound by Hb in NB5-6 are enriched for Gsb occupancy, although we note that occupancy may occur at different times (Gsb earlier, Hb later).

Taken together, these data support the sequential specification model, where a transiently expressed STF (e.g. Gsb) sculpts a lineage-specific chromatin landscape in NB lineages (eg. NB5-6), this determines lineage-specific binding of TTFs (e.g. Hb), which can in turn specify different neural fates in different NB lineages (*Figure 8*).

## Discussion

Since its first report, Targeted DamID has been used in multiple cell types, in both Drosophila and mammalian embryonic stem cells (ESCs), for mapping transcription factor binding (*Cheetham et al., 2018*; *Tosti et al., 2018*), open chromatin domains (*Aughey et al., 2018*), chromatin states (*Bonneaud et al., 2017*), and for mapping paused or transcribed loci (*Southall et al., 2013*; *Widmer et al., 2018*). In all cases, the number of cells expressing the Dam constructs are relatively large:~10,000 FACS purified ESCs (*Cheetham et al., 2018*) and ~5000 mushroom body neurons per brain (*Widmer et al., 2018*). In our study we analyze the smallest percentage of cells to date - we calculate that there are between 8–12 cells in each hemisegment expressing Dam constructs; with a total of 11 segments that would give a maximum of 264 cells per embryo, or about 0.5% of the estimated 50,000 cells per embryo. Furthermore, we pushed the limits of the technique by allowing just 5 hr of Dam or Dam:Hb expression. It's likely that this restrictive condition was successful in the case of a transcription factor-DNA interaction, which is stable during the time window; it might not be sufficient for factors such as RNA Pol II that require processivity through a gene. The ability to query transcription factor occupancy in such a precise manner – in a small subsets of cells over short

periods of time – will encourage new uses of the method, such as studying the determination of cellular identities during development, upon reprogramming, or even in response to stimuli.

We propose that the spatial factor Gsb opens genomic loci in NB5-6, allowing the temporal factor Hb to bind loci that are not available in the adjacent Gsb-negative NB7-4. Although nothing is currently known about the role of Gsb in chromatin regulation, the closely related mammalian Pax3 and Pax7 transcription factors can recruit histone methyltransferase to promote open chromatin and increase gene expression (*McKinnell et al., 2008*; *Diao et al., 2012*; *Kawabe et al., 2012*). Moreover, Pax7 is a pioneer factor during pituitary development, opening ~2500 loci (*Budry et al., 2012*). It would be informative to test whether Gsb can recruit trithorax complex methyltransferase to open genomic loci in row five neuroblasts, and whether this is required for row five neuroblast spatial identity and differential binding of Hb.

The specific enrichment of Gsb occupancy at regions of accessible chromatin in NB5-6 is a striking result that supports our model despite different cell populations used for each experiment (total embryonic vs. single NB lineage), different stages assayed (0–12 vs. 9–12), and different methods used (Dam vs. Gsb ChIP). Despite these differences, we observed significant enrichment of Gsb-bound loci at open chromatin in a NB-specific manner: NB5-6 shows enrichment, whereas NB7-4 does not. Ideally, similar experiments need to be conducted with Dam:Gsb in NB5-6 and Dam:En in NB7-4 lineage to determine correspondence of STF occupancy and chromatin accessibility, as well as STF and TTF occupancy in the NB lineages. The advantage of the Drosophila model is that these relationships can be rigorously tested. For example, mutational inactivation of the relevant STF, while assaying chromatin accessibility or Hb occupancy in a lineage-specific way could reveal a causal link between the STF and chromatin landscape, and STF and Hb occupancy. Similarly, targeting chromatin modifiers to select loci while assaying Hb occupancy could demonstrate a causal link between chromatin state and Hb occupancy. To definitively rule out the possibility that Hb acts as a pioneer in these lineages, it may be feasible to misexpress or mutate Hb, to determine the effect on chromatin accessibility. These are technically difficult studies, beyond the scope of this paper.

We show that ~1200 Hb-bound loci are different in NB5-6 and NB7-4 lineages, and that the chromatin at these sites is preferentially open. In some cases Dam:Hb occupancy is broader than Dam (open chromatin) occupancy; this could be due to Dam:Hb maintaining occupancy longer than Dam alone. The strong correlation between Dam:Hb binding and open chromatin could be due to Hb binding to previously opened chromatin domains, or Hb acting as a pioneer factor to open chromatin. We do not favor the latter mechanism because Hb binds some sites in NB5-6 but not in NB7-4 (and vice versa) showing that it is not sufficient to open chromatin.

NB5-6 and NB7-4 develop adjacent to each other during neuroblast formation. They share a common lateral Msh+ spatial column, but are in different anterior/posterior spatial domains (NB5-6 is Gsb+, NB7-4 is En+). Although NB5-6 and NB7-4 make different early-born neurons, they share a common ability to make subperineurial glia and neurons that project through the posterior commissure (*Schmidt et al., 1997*; *Schmid et al., 1999*). It is interesting to speculate that their common properties are due to their

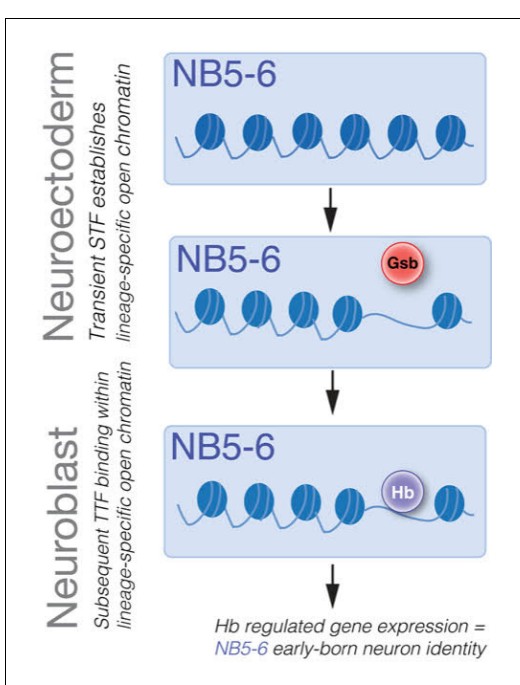

**Figure 8.** Sequential specification integrates spatial and temporal cues to generate diversity in Drosophila embryonic NB lineages. Transient expression of spatial factors in the neuroectoderm (e.g. Gsb in row 5) establishes lineage-specific chromatin landscapes (e.g. NB5-6 lineage). Subsequently, TTFs (e.g. Hb) in the NB can access different genomic targets to regulate different genes in spatially distinct NB lineages. This results in the specification of different neural fates in different NB lineages.
DOI: https://doi.org/10.7554/eLife.44036.017

shared columnar spatial position, whereas their differences are due to different anterior/posterior spatial cues.

Although we have provided evidence that Hb-bound loci are chosen from neuroblast-specific open chromatin domains, this does not rule out that sequential specification occurs via lineage-specific STFs/STF-target genes acting as Hb cofactors to bias Hb binding in each lineage. However, we have been unable to find any *de novo* DNA motif enriched within 1 kb of Hb-bound loci throughout the genome, either neuroblast-specific loci or within all Hb-bound loci. This is consistent with Hb acting independently, but we can't rule out the possibility of Hb acting with co-factors. Our conclusions are in agreement with studies showing that DNA accessibility, not cooperative or competitive interactions, have the strongest impact on transcription factor binding (*Li et al., 2008*; *Kaplan et al., 2011*). Similarly, this model is supported by in vitro protein-DNA studies that eliminate chromatin state contribution to these interactions (*Guertin et al., 2012*).

Using traditional methods of studying protein-DNA interactions, Hb targets in early embryogenesis have been well-characterized (*Hoch et al., 1991*; *Struhl et al., 1992*; *Rivera-Pomar et al., 1995*; *Berman et al., 2002*), yet little is known about Hb direct targets in the CNS, and nothing is known about neuroblast lineage-specific targets that specify lineage-specific neuronal identity. Here we've reported the first description of Hb occupancy in vivo within the genome of individual neuroblast lineages. Our study identified many loci that were similarly occupied in the two lineages, which are likely to consist of regulatory modules common to both lineages such as pan-neuronal specification or the progression of the temporal series. The latter example consists of Hb activating *Kr* and repressing *pdm2* in most neuroblast lineages. Indeed we find that Hb binds to both loci in NB5-6 and NB7-4 lineages, confirming previous observations that Hb directly represses *pdm2* and activates *Kr* in multiple neuroblast lineages (*Kambadur et al., 1998*; *Tran et al., 2010*). Hb is also likely to directly repress *zfh2* in most neuroblast lineages (CQD, unpublished results) and our data show that the *zfh2* locus is indeed equivalently occupied in both neuroblast lineages. Apart from the commonly regulated loci, we identified over 100 loci that are differentially bound by Hb in NB5-6 or NB7-4. These are excellent candidates for lineage-specific neuronal specification.

Our study, coming almost two decades after the first descriptions of spatial and temporal patterning in Drosophila neural stem cells (*Isshiki et al., 2001*), has for the first time explored the mechanism by which spatial and temporal factors could be integrated to generate neuroblast-specific neuronal progeny. Only recently has it been possible to probe TTF DNA-binding and chromatin landscapes within two distinct neuroblast lineages – due to the parallel advances in genetic tools, functional genomics, and our ability to manipulate the genome. Given the conservation of mechanisms in generating neural diversity in vertebrates and invertebrates, and exquisite ways in which the genome can now be manipulated in different organisms, it is now possible to determine if similar mechanisms generate diversity during vertebrate neurogenesis.

## Materials and methods

**Key resources table**

| Reagent type (species) or resource | Designation | Source or reference | Identifiers | Additional information |
|---|---|---|---|---|
| Strain, strain background (*Drosophila melanogaster*) | UAS-LT3-Dam | A Brand | NA | UAS drives mCherry from 1st cistron and Dam from 2nd cistron |
| Strain, strain background (*D. melanogaster*) | engrailed-Gal4 | A Brand | NA | Expressed in row 6 and 7 neuroblasts |
| Strain, strain background (*D. melanogaster*) | R19B03[AD]; R18F07[DBD] | G Rubin | NA | Expression in NB7-4 lineage from stage 9 |

*Continued on next page*

*Continued*

| Reagent type (species) or resource | Designation | Source or reference | Identifiers | Additional information |
|---|---|---|---|---|
| Strain, strain background (*D. melanogaster*) | Lbe(K)-Gal4 | S Thor | NA | Expression in NB5-6 lineage from stage 9; salivary gland at stage 17 |
| Strain, strain background (*D. melanogaster*) | UAS-LT3-Dam:Hb | This paper | NA | UAS drives mCherry from 1st cistron and Dam:Hb from 2nd cistron |
| Strain, strain background (*D. melanogaster*) | Sca-Gal4 | Y Hiromi | NA | Expressed in all NBs |
| Strain, strain background (*D. melanogaster*) | hsFLP;; UAS-MCFO | A Nern | NA | MCFO (multi-colored-flip-out) line |
| Strain, strain background (*D. melanogaster*) | MARCM stock | T Lee | NA | For clonal analysis of NB7-4 lineage |
| Strain, strain background (*D. melanogaster*) | UAS-HA:UPRT | Doe lab | NA | Control transgene |
| Strain, strain background (*D. melanogaster*) | Da-Gal4 | BDSC | 55850 homozygous on III | |
| Antibody | chicken anti-GFP (polyclonal) | Abcam (Eugene, OR) | ab13970 | (1:1000) |
| Antibody | mouse anti-en (monoclonal) | DSHB (Iowa City, IA) 4D9 | | (1:50) |
| Antibody | rabbit anti-Dan (polyclonal) | Doe lab | NA | (1:1000) |
| Antibody | rabbit anti-Hb (polyclonal) | Doe lab | NA | (1:400) |
| Antibody | rabbit anti-Eve (polyclonal) | Doe lab | SC1320A | (1:500) |
| Antibody | rat anti-Gsb (monoclonal) | Holmgren Lab | 1:1 10E10/16F2 | (1:10) |
| Antibody | mouse anti-mCherry (polyclonal) | Clonetech | 632543 | (1:500) |
| Antibody | rabbit anti-V5::549 (polyclonal) | Rockland | 600-442-378 | (1:400) |
| Antibody | mouse anti-HA::488 (monoclonal) | Cell signaling | 2350S | (1:200) |
| Antibody | rat anti-Ollas::650 (monoclonal) | Novus | NBP1-06713 | (1:200) |
| Antibody | Secondary antibodies (polyclonal) | Thermofisher (Eugene, OR) | | (1:400) |

## Fly lines

Fly stocks were obtained from the Bloomington Drosophila Stock Center (Bloomington, IN USA) and, unless otherwise stated, were grown on cornmeal media at 25°C. *UAS-LT3-Dam* flies were kindly provided by Andrea Brand, *R19B03[AD]; R18F07[DBD]* was a gift from Gerald Rubin, and *Lbe-(K)-Gal4* (called *NB5-6-Gal4* here) was a gift from Stephan Thor. To generate MCFO clones (*Nern et al., 2015*) with *NB5-6-Gal4* or *NB7-4-Gal4*, we crossed *hsFLP; UAS-MCFO* females to Gal4 line males. 0–1 hr eggs were collected, aged at 25C until stage eight and given a 37°C heat shock for 20 min then aged at 25°C or 18°C until stage 17. We used MARCM (*Lee and Luo, 1999*) with *engrailed-Gal4* to generate NB7-4 clones, which were unambiguously identified by the presence of channel glia (*Schmidt et al., 1997*; *Schmid et al., 1999*).

## Immunohistochemistry and confocal imaging

Embryos were dechorionated in bleach for 3 min and fixed in 1:1::4% PFA:Heptane for 20–30 min. Vitteline menbranes were removed by shaking them vigorously in 1:1::heptane:methanol. They were washed with blocking solution (1 × PBS with 0.3% TritonX and 0.1% BSA) for an hour. Primary antibodies were diluted in blocking solution. The samples were incubated on horizontal shaker at 4°C for 24 hr after which they were washed with 0.3% PTX (1 × PBS with 0.3% TritonX) and secondary antibody diluted in 0.3% PTX was added. The samples were incubated at 4°C overnight, washed 0.3% PTX, allowed to settle in 30% glycerol, then allowed to clear in 90% glycerol infused with Vectashield overnight. Primary antibodies used were: chicken anti-GFP (1:1000, abcam ab13970), mouse anti-engrailed (1:50, 4D9 DSHB); rat anti-gooseberry (1:10 of equal mix of 10E10 and 16F2, Holmgren Lab), rabbit anti-Hunchback (1:400), rabbit anti-Dan (1:1000), mouse anti mCherry (1:500, Clonetech 632543), rabbit anti-V5::549 (1:400, Rockland 600-442-378), mouse anti-HA::488 (1:200, Cell signaling 2350S), rat anti-Ollas::650 (1:200, Novus NBP1-06713) and rabbit anti-Eve (1:500). All samples were imaged on ZeissLSM700 or ZeissLSM710 confocal microscope. Optical sections were acquired at 0.75 µm intervals with a picture size of 1024 × 1024 pixels. Images were processed in the open source software FIJI (http://fiji.sc).

## Generation of Dam:Hb

To generate *UAS-LT3-Dam:hb*, full-length *hb* CDS was PCR amplified from BACR01F13 and cloned into *pUAST-attB-LT3-NDam* (a gift from Andrea Brand) using NotI and XbaI sites to fuse Dam to the N-terminus of Hb. As spontaneous mutations are known to arise in the Dam sequence upon transformation (*Marshall et al., 2016*), its sequence integrity was tested at each transformation step, and prior to injections, all three elements - Dam, Hb and Cherry sequences were confirmed to be preserved. Transgenic flies with the construct integrated at the attP2 landing site were generated by BestGene Inc.

## Dam:Hb and Dam genomic binding

For verifying the Dam:Hb flies, about 1500 females of *UAS-LT3-Dam* and *UAS-LT3-Dam:hb* flies were crossed to about 500 males of *Da-Gal4* in egg collection cages placed at 25°C. Embryos were collected every two hours and aged for 16 hr at 25°C, then dechorionated with bleach to avoid contaminants, washed thoroughly with de-ionized water and preserved at −20°C until sufficient material was collected - for each replicate, 50 mg of control and experimental embryos. For stage 12 neuroblast TaDa experiments, about 5,000–6,000 *UAS-LT3-Dam* and *UAS-LT3-Dam:hb* flies were crossed to about 3,000 *Lbe-K-Gal4* or *19B03[AD]/18F07[DBD]* flies. Embryos were collected every two hours and aged for 7.5 hr at 25°C, and similarly treated until sufficient material was collected - for each replicate, 4 × 1.5 µL tubes of 50 mg of control and experimental embryos.

The TaDa experimental pipeline was followed according to *Marshall et al. (2016)*, with a few alterations to optimize for small cell numbers and short duration of Dam expression. Briefly, the 4 tubes of each replicate were thawed on ice, processed separately and in parallel until the PCR purification step after the DpnI digestion step; subsequently, an additional PCR purification step using standard Qiagen PCR purification columns was used to concentrate the DpnI digested product to 32 µL. Embryos were homogenized with an electric pestle and gDNA was extracted using the DNA Micro Kit (Qiagen, cat. no. 56304). Extreme care was taken to ensure that the gDNA remained intact – this was done by using wide bore tips to avoid fragmenting the DNA, pipetting deliberately, and

avoiding any rough shaking/tipping. gDNA was digested with DpnI for 14–16 hr in a thermocycler then PCR purified. MyTaq HS DNA polymerase kit (Bioline, cat. no. BIO-21112; not the Advantage 2 cDNA polymerase from Clonetech) was used for amplification and 21 PCR cycles we used. Sequencing libraries were prepared according to the Illumina TruSeq DNA library protocol. The samples were sequenced on the Illumina HiSeq4000 at 100 base pairs and about 20–60 million single end reads per sample.

## Bioinformatic analysis

### Quality control
Each file was assessed for quality using FastQC (*Andrews, 2010*). Reads with quality score less than 30 were discarded. Any contaminants were removed using BBsplit of the BBmap suite (https://sourceforge.net/projects/bbmap/ ).

The damidseq_pipeline was used to generate log2 ratio files (Dam:hb/Dam) in GATC resolution as described previously (*Marshall and Brand, 2015*). Briefly, the pipeline uses Bowtie2 (*Langmead and Salzberg, 2012*) to align reads to dm6, the reads are extended to 300 bp (or to the closest GATC, whichever is first) and this .bam output is used to generate the ratio file (.bedgraph). Normalization: reads are sorted into deciles. The top decile in the Hb:Dam fusion, and the bottom three deciles from the Dam alone are excluded from the normalization to avoid loss of true signal and reduce noise respectively. A normalization factor is calculated on the log2 ratio of the remaining reads. For more details on the DamID-seq pipeline and normalization process, please see *Marshall and Brand (2015)*.The bedgraph files were used for data visualization on IGV 2.4.1 (*Robinson et al., 2011*; *Thorvaldsdóttir et al., 2013*) and the read extended bam files were used for peak calling.

Correlation coefficients between biological replicates for Da-Gal4 Hb TaDa and Da-Gal4 CaTaDa were computed using the multiBamSummary and plotCorrelation functions of DeepTools. For NB5-6 and NB7-4 Hb TaDa and CaTaDa, where differential analyses were conducted, the correlation coefficients computed by DiffBind (*Ross-Innes et al., 2012*) are represented.

### Peak calling
For TaDa experiments, MACS2 (v2.1.1) (*Zhang et al., 2008*) was used to call narrow peaks on sorted, read extended bam files of Dam:Hb, with a single merged Dam only as a control provided for each replicate. MACS2 (v2.1.1) was also used to call peaks on Hb ChIP-seq data. For this, dm3 aligned Hb ChIP-seq and input files (in bowtie output format) were downloaded from NCBI (GEO accession number GSE20369; HB2) and converted to sam format using bowtie2sam.pl from the SAMtools suite. These were converted to bam and CrossMap (*Zhao et al., 2014*) was then used to liftOver both the input and Hb files from dm3- > dm6. deepTools was used to generate the ratio files for subsequent analyses. For CaTaDa experiments, narrow peaks were called on sorted, read extended bam files of Dam only using MACS2 (v2.1.1) without controls.

### Peak overlap
Bedtools intersect was used for computing peak overlaps. An overlap of 1 basepair or more was considered an overlap. *Hb ChIP-seq vs. Hb TaDa:* narrow peak output from MACS2 were used for both files. *Da-Gal4 CaTaDa vs. DNAseI*: the MACS2 generated narrow peaks for *Da-Gal4* CaTaDa was supplied along with the stage 11 DNAseI peak file, which was downloaded from BDTNP and lifted over from dm2- > dm6 using CrossMap. *Differential Hb vs. differential chromatin*: the differentially bound sites identified by DiffBind (*Ross-Innes et al., 2012*) were saved as bed files and provided to bedtools intersect to assess overlap percentage. *Differential Hb vs.Gsb.* bedtools closest was used to detect the closest Gsb peak to the peak centres of NB5-6 and NB7-4 Hb enriched regions. Fishers test was performed using bedtools fisher.

### Monte Carlo analysis
To check for the significance of peak/signal overlap, a Monte Carlo analysis was performed. *Hb TaDa vs. Hb ChIP:* Hb ChIP was taken as the reference, and an equal number of random peaks were generated such that the number and length of peaks for each chromosome remained the same. These random peaks were used to check for overlap with Hb TaDa. A 100 such iterations were

performed, and an average overlap calculated for the random overlap. Z-score and p-value was calculated between the average random overlap and the actual overlap. A custom written script was used to perform this analysis (*Aughey et al., 2018*). *Da-Gal4 CaTaDa vs. DNAseI*: Similar analysis as above was used with DNAseI as the reference. *Differential Hb and Differential chromatin*: Differentially bound, thresholded Hb peaks of NB5-6 and NB7-4 were taken as the reference and an equal number of random peaks were generated such that the number and length of peaks for each chromosome remained the same. These random peaks were used to check for overlap with the differentially bound chromatin loci in the respective NB. A 100 such iterations were performed, and an average overlap calculated for the random overlap. The Z-score and p-value were calculated between the average random overlap and the actual overlap. *Gsb signal at 5–6 and 7–4 chromatin and enriched Hb loci*: 'bedtools slop' was used to extend the 5–6 and 7–4 peaksets to 4 kb (2 kb on either side of the peak center). An equal number of random peaks were generated for 5–6 and 7–4 as in the actual data (respecting distribution of peaks on the chromosomes). 'bedtools shuffle' was used to generate these random peaks. The Gsb data obtained from Florence Maschat was converted from wig to bedgraph using 'wig2bed' from bedops, then dm3- > dm6 using CrossMap, and finally from bedgraph to bigwig using 'bedGraphToBigWig' from kentUtils (https://github.com/ENCODE-DCC/kentUtils). 'bigWigAverageOverBed' from kentUtils was used to generate the average Gsb signal at each peak. The average signal for each iteration was generated using awk. The difference in average Gsb signal between (randomly generated) NB5-6 and (randomly generated) NB7-4 was calculated for a 1000 such iterations. The difference between average Gsb signal for the real data (i.e. 5–6 enriched Hb loci minus 7–4 enriched Hb loci) was similarly calculated. Z scores and p-values were calculated based on these 1000 simulations and real differences in Gsb signal. A bash script was written to automate the above steps (available upon request). Similar pipeline was used for comparisons with *bcd*, *kni*, *cad* and *Kr*.

## TaDa/CaTaDa signal comparisons with other data

The computeMatrix tool from deepTools was used to plot the signal distribution relative to reference points in *Figure 3F,G*; 5B; 6A-C; 7A,B; and *Figure 6—figure supplement 1*. In all cases, signal files (of ChIP or TaDa data) were supplied as bigwig files, and peaks regions were supplied as bed files. *Figure 3F* peak file was the narrow peaks generated by MACS2 in the three Da-Gal4 Hb TaDa experiments; the Hb ChIP-seq ratio file was used as the signal file (see under peak calling for details). *Figure 3G* peak files for Hb, Bcd and Ftz were downloaded from BDTNP and were lifted-over from dm3- > dm6 using CrossMap; the Hb TaDa signal was converted to bigwig using 'bedGraphToBigWig' from kentUtils (https://github.com/ENCODE-DCC/kentUtils). *Figure 5B* peak file was downloaded from BDTNP and was lifted-over from dm2- > dm6 using CrossMap; the Da-Gal4 CaTaDa signal was converted to bigwig using 'bedGraphToBigWig' from kentUtils. *Figure 6A–C*: separate region files were made from the DiffBind (*Ross-Innes et al., 2012*) output for NB5-6 enriched, 7–4 enriched and 'Not-Differentially Bound' Hb loci; NB5-6 and NB7-4 CaTaDa files were converted to bigwig using 'bamCoverage' of deepTools. *Figure 6—figure supplement 1A,B*: MACS2 generated narrow peaks for NB5-6 and NB7-4 were used; NB5-6 and NB7-4 CaTaDa files were converted to bigwig using 'bamCoverage' of deepTools. *Figure 7A*: All MACS2 generated narrow peaks on the NB5-6 and NB7-4 CaTaDa were supplied as the regions of open chromatin; Gsb ChIP-chip signal file was used (see under Monte Carlo analysis for details). *Figure 7B*: separate region files were made from the DiffBind (*Ross-Innes et al., 2012*) output for NB5-6 enriched and 7–4 enriched Hb loci; Gsb ChIP-chip signal file was used (see under Monte Carlo analysis for details).

Motif calling was performed using the findMotifs.pl tool from the Homer suite of tools. The top 1000 narrow peaks from MACS2 were supplied to Homer and de novo motif calling was performed on 300 kb on either side of the peak centre. Approximately 6.5 times the number of supplied peaks were used as background to calculate enrichment. Using all peaks gave comparable results, with Hb as the most enriched motif over background.

Differential analyses in *Figure 4* and *Figure 5* were performed using DiffBind (*Ross-Innes et al., 2012*). Briefly, narrow peak output files were provided for each of the three replicates of NB5-6 and NB7-4, along with their aligned Dam:Hb (*Figure 4*) or Dam alone (*Figure 5*) bam files. An initial correlation was calculated between the samples (both between replicates and across NBs) at these loci. The number of overlapping reads at each region was calculated, normalized, and represented as a

*binding affinity matrix.* This matrix data was used for the further differential binding analysis and assignment of FDR and p-values, which can be conducted using either DeSeq2 or edgeR packages. Data shown here are results from DeSeq2 based differential analyses. Correlation heatmap, binding affinity matrix, MA plots and volcano plots represented in *Figure 4* and *Figure 5* were generated using Diffbind (*Ross-Innes et al., 2012*).

## Acknowledgements

We thank Keiko Hirono and Dylan Heussman for generating the Dam:Hb transgene; Sen-Lin Lai for *Figure 2J*; Keiko Hirono for contributing to *Figure 3A*; Andrea Brand for TaDa reagents; Stephan Thor for Lbe reagents; Gerry Rubin for 7–4 Gal4; Jan Trout for *Figure 1* illustrations; and Maggie Weitzman and Douglas Turnbull at the UO Genomics facility. We thank Sen-Lin Lai, Brandon Mark, Heinrich Reichert, Vishaka Datta, Gabriel Aughey, and Richard Mann for comments on the manuscript. Stocks obtained from the Bloomington Drosophila Stock Center (NIH P40OD018537) were used in this study. Funding was provided by the Fulbright-Nehru Postdoctoral fellowship (SQS), HHMI (CQD, SQS, SC), and NIH HD27056 (CQD).

## Additional information

### Funding

| Funder | Grant reference number | Author |
| --- | --- | --- |
| Howard Hughes Medical Institute | | Sonia Q Sen<br>Sachin Chanchani<br>Chris Q Doe |
| Fulbright-Nehru Postdoctoral Fellowship | | Sonia Q Sen |
| National Institutes of Health | HD27056 | Chris Q Doe |

The funders had no role in study design, data collection and interpretation, or the decision to submit the work for publication.

### Author contributions

Sonia Q Sen, Conceptualization, Formal analysis, Supervision, Validation, Investigation, Visualization, Methodology, Writing—original draft, Writing—review and editing; Sachin Chanchani, Resources, Data curation, Software, Investigation, Methodology; Tony D Southall, Resources, Formal analysis, Investigation, Methodology, Writing—review and editing; Chris Q Doe, Conceptualization, Data curation, Supervision, Funding acquisition, Writing—original draft, Project administration, Writing—review and editing

### Author ORCIDs

Sonia Q Sen (iD) http://orcid.org/0000-0003-4693-3378
Tony D Southall (iD) http://orcid.org/0000-0002-8645-4198
Chris Q Doe (iD) http://orcid.org/0000-0001-5980-8029

### Decision letter and Author response

Decision letter https://doi.org/10.7554/eLife.44036.025
Author response https://doi.org/10.7554/eLife.44036.026

## Additional files

### Supplementary files

• Supplementary file 1. Hb enriched loci in NB5-6 lineage and NB7-4 lineages. The genome coordinates for Hb enriched loci in NB5-6 and NB7-4. Columns show, from left to right: chromosome arm, start of fragment, end of fragment, concentration, concentration in NB5-6, concentration in NB7-4, fold enrichment (NB5-6/NB7-4), p-value, and False Discovery Rate (FDR).

DOI: https://doi.org/10.7554/eLife.44036.018
• Transparent reporting form
DOI: https://doi.org/10.7554/eLife.44036.019

## Data availability

Data are available via the NCBI Gene Expression Omnibus database (accession number GSE123272).

The following dataset was generated:

| Author(s) | Year | Dataset title | Dataset URL | Database and Identifier |
|---|---|---|---|---|
| Chris Q Doe, Sonia Q Sen | 2019 | Neuroblast-specific open chromatin allows the temporal transcription factor, Hunchback, to bind neuroblast-specific loci | https://www.ncbi.nlm.nih.gov/geo/query/acc.cgi?acc=GSE123272 | NCBI Gene Expression Omnibus, GSE123272 |

The following previously published dataset was used:

| Author(s) | Year | Dataset title | Dataset URL | Database and Identifier |
|---|---|---|---|---|
| Bradley RK, Li XY, Trapnell C, Davidson S | 2010 | Binding site turnover produces pervasive quantitative changes in TF binding between closely related Drosophila species | https://www.ncbi.nlm.nih.gov/geo/query/acc.cgi?acc=GSE20369 | NCBI Gene Expression Omnibus, GSE20369 |

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
