## [Decision Letter]

Thank you for submitting your article "Neuroblast-specific chromatin landscapes allow integration of spatial and temporal cues to generate neuronal diversity" for consideration by *eLife*. Your article has been reviewed by three peer reviewers, including Gail Mandel as the Reviewing Editor and Reviewer #1, and the evaluation has been overseen by Kevin Struhl as the Senior Editor. The following individuals involved in review of your submission have agreed to reveal their identity: Claude Desplan (Reviewer #2); Michael B Eisen (Reviewer #3).

The reviewers have discussed the reviews with one another and the Reviewing Editor has drafted this decision to help you prepare a revised submission.

This work describes an elegant application of a recently developed technique (targeted DAMID) to determine chromatin changes mediated by transcription factors that regulate spatiotemporal gene expression in *Drosophila* neuroblasts. Authors provide results suggesting that the chromatin changes mediated by the spatial factors establish a permissive environment for activity of the temporal factors important in lineage control (intersection of spatial and temporal identify mechanisms).

There were few major concerns identified by all three reviewers, who were very enthusiastic about the novelty and rigor in the technique application and the importance of the question. These concerns are easily addressed and don't require further experimentation (please see attached reviews). However, one concern that merits more attention is that the data in Figure 7, the crux of the conclusion for integration of the chromatin signaling, does not appear to be to the same level of rigor as the technical aspects. Authors need to address this point, by providing either more convincing data in Figure 7 or, minimally, providing more details in the results and interpretation of Figure 7 data, as well as toning down wording in the title, Abstract and Discussion that they have proven intersection, as opposed to generating data that is consistent with this conclusion.

Essential revisions: Figure 7 data and toning down language if no more data is provided.

*Reviewer #1:*

This work is meant to address the intersection of temporal and spatial information leading to neuronal diversity (distinct neuroblast lineages), an intersection that has received very little traction. The work takes advantage of the huge breadth of knowledge of the spatial transcription factors (STFs) and hunchback transcription factor (TFF) cascades for neuronal lineage control in *Drosophila*. The study also applies a new method of binding site identity (DAMID) that has not been applied previously for small numbers of cells or for addressing this specific question. Applying this method, apart from the question of integration signaling, the authors have identified 100 new targets that could potentially contribute importantly to neuronal specification.

Shown in an elegant manner is that STF Gsb and DamHb bind within open chromatin, defined by DAM analysis, in a neuroblast-specific manner. Because Gsb binds prior to Hb, authors propose a sequential model wherein open chromatin induced by Gsb binding is required for subsequent binding of Hb. However, their data in support of this model (Figure 7) falls short of showing causality and doesn't seem to be done at the same level of rigor as in the prior experiments, leading to somewhat of an anticlimax. For example, there is no direct evidence presented that the STF open chromatin is sufficient for binding of Hb, only that the binding of the two factors is enriched in close proximity in open chromatin. Additionally, the data in Figure 7B is not completely convincing – the binding enrichment curves are quite broad and appear very noisy, suggesting that the n value of # peaks is very small, although the Monte Carlo analysis shows significance (throughout the figures authors should provide an n value in their plots). While I think the work represents a clever adaptation of the technique, rigor for establishing the technique, first demonstration of in vivo binding of Hb, identification of potentially new factors important in specification, and setting the stage for attacking an important unanswered question, I think the question is still, well, an open question. In their Discussion, authors indicate that experiments to determine causality of Gsb binding/open chromatin for Hb binding lie outside the scope of the paper. Agreed, such studies would involve further work, but as it stands the current study doesn't support the bold title that the chromatin landscape allows integration. The Abstract wording is more accurate, but saying in the Impact statement and Introduction that the integration is due to and support (as opposed to consistent with) the sequential model, and asking whether similar mechanisms occur in vertebrates, seems overstated based on the current data.

Unless I missed it, authors do not state explicitly precisely how close the Hb and STF sites of enrichment are? Related to this, in terms of strengthening the correlative data, authors might consider plotting the distributions of distances of the closest Gsb peaks (or motifs) from the peak center of the Dam:Hb peaks and doing the same for other "control" STF or TFFs/motifs Chip data. Authors indicate that they didn't see any other motifs close to Hb sites but it wasn't clear whether the analysis was genome wide? It might also be optimal for authors to perform their own ChIP experiments to make this critical point. Regarding the Discussion. How does Gsb open chromatin – must be recruiting enzymes? Anything known about a Gsb complex? Are the Gsb binding sites associated with enhancer chromatin marks?

*Reviewer #2:*

This a very carefully crafted manuscript that analyzes how spatial and temporal information are integrated in neural stem cells to generate the large diversity of neurons in the ventral nerve cord of *Drosophila*.

The authors wanted to assay the mechanisms of molecular integration between the two sets of transcription factors (TFs).

They chose to look at the binding sites for the best known temporal TF, Hunchback (Hb) in two spatially distinct neuroblasts. Because of the very small number of neurons available in each embryo, the authors chose to use a very clever method initially developed in Andrea Brand's lab, TaDa. This method relies on the specific expression of a Dam methylase fused to the TF to test in specific cell types, but requires difficult adjustments as Dam can be very toxic even at low concentrations.

What makes this paper special is the very careful evaluation of the Gal4 lines used to mark two specific neuronal lineages, but more importantly the evaluation of how Dam-Hb functions. This allowed the authors to evaluate the differential binding of Hb to its targets in the two lineages. These differ significantly, suggesting that spatial information instructs the ability of Hb to bind to its (important) targets, likely through opening of chromatin, which they tested.

For this purpose, they also used Dam without its targeting moiety: even with the minuscule number of cells, they could see that there is a correlation between differential open chromatin and Hb binding. It is quite amazing that they got this to work but the controls appear to be fine. If this works that well, others should use this approach before single cell ATACseq becomes available!

The spatial transcription factor Gsb expressed early in the lineage appears to be responsible for this opening.

In conclusion, the use of the very powerful and highly focused TaDa technique allowed the authors to propose a model where chromatin is differentially opened by spatial TFs which allow the same temporal TFs to define distinct lineages. I am impressed by the technical sophistication of the paper and the care with which this has been done, which led to this important conclusion.

Of course, I would have liked to see other spatial and other temporal TFs being tested but in keeping with the spirit of *eLife*, I think that the paper makes an important enough contribution to be published without much change.

*Reviewer #3:*

The meat of this paper is the use of cell-type specific DamID to compare Hunchback (Hb) binding in two populations of neuroblasts distinguished by the expression of different spatial transcription that both respond to a pulse of Hb expression to make distinct neurons. The authors establish through a set of control experiments and comparisons to other data the efficacy of using specific expression of Dam:Hb to identify Hb target sites, and the viability of the neuroblast specifically expression Dam:Hb using cell-type specific drivers. The results are pretty straightforward: Hb binds to different targets in these two neuroblast subpopulations. They then show that this differential binding corresponds to differential chromatin accessibility, leading to their primary conclusion, that the differential binding of Hb (and presumably other temporal transcription factors) is due to the establishment of distinct chromatin states. They present data suggesting that the spatial transcription factors Gsb might be responsible for establishing these differential states in one subpopulation, lending support for a general model for neuroblast specification in which spatial transcription factors create a unique chromatin state that shapes how temporal transcription factors create identity.

I found the data generally compelling and don't have any major issues. Of course this is just binding, measured indirectly with a technique that whose pitfalls are not well established, and the evidence for STF involvement in establishing chromatin states is based on one factor. But as a first pass it's good data of great interest that warrants publication.

One thing confused me. The Abstract says:

"Profiling chromatin accessibility showed that each neuroblast had a distinct chromatin landscape: Hunchback-bound loci in NB5-6 were in open chromatin, but the same loci in NB7-4 were in closed chromatin."

I assume this is just poorly worded since it seems to contradict what's said in the paper (The data show that Hb binding in NB7-4 is in open chromatin in NB7-4)? I'm putting this in the major comments section since having an Abstract that says the opposite of the paper isn't good.

---

## [Author Response]

[…] However, one concern that merits more attention is that the data in Figure 7, the crux of the conclusion for integration of the chromatin signaling, does not appear to be to the same level of rigor as the technical aspects. Authors need to address this point, by providing either more convincing data in Figure 7 or, minimally, providing more details in the results and interpretation of Figure 7 data.

Reviewer #1:[…] Additionally, the data in Figure 7B is not completely convincing – the binding enrichment curves are quite broad and appear very noisy, suggesting that the n value of # peaks is very small, although the Monte Carlo analysis shows significance (throughout the figures authors should provide an n value in their plots).

We appreciate these comments, and have added new text to the Discussion and a new panel to Figure 7. See also new data in response to the third comment below. There are several likely reasons for the relatively low (but significant!) correlation between Gsb occupancy and the open chromatin states of the two NBs. First, different cell populations are used (NB lineages vs. total embryonic cells), different stages are assayed (0-12 vs. 9-12), different methods are used (ChIP vs. Dam). Despite these differences we were actually very pleasantly surprised to see significant enrichment of Gsb bound loci at open chromatin in a NB-specific manner – NB 5-6 shows enrichment, whereas NB7-4 does not. We have added the above text to the Discussion. We have also added a graphical representation of the Monte Carlo analysis used in 7B to the revised Figure 7 (new 7C), which demonstrates the significance of the enriched Gsb binding at unique NB5-6 Hb peaks.

We agree that it would be ideal to compare Dam (open chromatin) to Gsb-Dam (Gsb binding), but we do not yet have a Gsb-Dam fly stock. We would be very interested in adding these data in the future as an *eLife* Research Advance (Patterson et al., 2014) linking back to our current paper.

Dr Mandel is also correct in noting that the number of peaks is small in 7B. While the number of peaks used in 7A (sites of open chromatin) are 20,838 and 18,201 for the NB5-6 and 29,817 and 31,080 for NB7-4, the number of peaks used in 7B (NB-specific Hb-bound loci) is 504 and 718. We have now mentioned the numbers of peaks used in these plots in the figure legend.

As well as toning down wording in the title, Abstract and Discussion that they have proven intersection, as opposed to generating data that is consistent with this conclusion. There is no direct evidence presented that the STF open chromatin is sufficient for binding of Hb, only that the binding of the two factors is enriched in close proximity in open chromatin. In their Discussion, authors indicate that experiments to determine causality of Gsb binding/open chromatin for Hb binding lie outside the scope of the paper. Agreed, such studies would involve further work, but as it stands the current study doesn't support the bold title that the chromatin landscape allows integration.

We appreciate this comment, and we have completely rewritten our title and Abstract accordingly. We have changed the title to: “Neuroblast-specific open chromatin landscapes allow the temporal transcription factor, Hunchback, to bind neuroblast-specific genomic loci” which leaves room for future experimental verification. In the Impact statement, Introduction, and Discussion we say that “our findings support a model” or “we propose that” in all places.

Unless I missed it, authors do not state explicitly precisely how close the Hb and STF sites of enrichment are? Related to this, in terms of strengthening the correlative data, authors might consider plotting the distributions of distances of the closest Gsb peaks (or motifs) from the peak center of the Dam:Hb peaks and doing the same for other "control" STF or TFFs/motifs Chip data.

This was a great suggestion, thank you! We found that of the 503 Hb enriched loci in NB5-6, 101 had a Gsb peak within 2Kb of the centre, whereas, this number was only 49 for NB7-4. A Fisher’s exact test on these data found this spatial relationship to be highly significant for NB5-6 (p=8.7812e-19), but not for NB7-4 (p=0.077982). These findings have been added to the text.

Authors indicate that they didn't see any other motifs close to Hb sites but it wasn't clear whether the analysis was genome wide? It might also be optimal for authors to perform their own ChIP experiments to make this critical point.

We now say in the last paragraph of the Discussion: “we have been unable to find any de novo DNA motif enriched within 1kb of Hb-bound loci throughout the genome.” We feel our Hb-Dam data is sufficient to identify Hb binding sites, and we have validated it against very high quality Hb ChIP experiments using stage 9 whole embryos with excellent correlation.

Regarding the Discussion. How does Gsb open chromatin – must be recruiting enzymes? Anything known about a Gsb complex? Are the Gsb binding sites associated with enhancer chromatin marks?

Thank you very much for this comment and for provoking us to dive into the mammalian Pax literature. Although *Drosophila* Gsb shows no protein or genetic interactions with chromatin regulators (Flybase and PubMed), its closest mammalian relatives, Pax3 and Pax7, are well-known to recruit trithorax complex proteins to open chromatin. We now cite these studies in the Discussion: “Although nothing is currently known about the role of Gsb in chromatin regulation, the closely related mammalian Pax3 and Pax7 transcription factors can recruit histone methyltransferase to promote open chromatin and increase target gene expression (Diao et al., 2012; Kawabe et al., 2012; McKinnell et al., 2008). […] It would be informative to test whether Gsb can recruit trithorax complex methyltransferases to open genomic loci in row 5 neuroblasts, and whether this is required for row 5 neuroblast spatial identity and differential binding of Hb.” These experiments are now among our highest priorities for the coming year!

Reviewer #3:[…] One thing confused me. The Abstract says:"Profiling chromatin accessibility showed that each neuroblast had a distinct chromatin landscape: Hunchback-bound loci in NB5-6 were in open chromatin, but the same loci in NB7-4 were in closed chromatin."I assume this is just poorly worded since it seems to contradict what's said in the paper (The data show that Hb binding in NB7-4 is in open chromatin in NB7-4)? I'm putting this in the major comments section since having an Abstract that says the opposite of the paper isn't good.

You are correct, it was poor wording. We have changed this sentence to say:

“each neuroblast had distinct open chromatin domains, which correlated with differential Hb-bound loci in each neuroblast.”